# The burden of disease and economic impact of sugar-sweetened beverages' consumption in Argentina: A modeling study

Ariel Esteban Bardach[1,2]*, Natalia Espínola[1], Federico Rodríguez Cairoli[1], Lucas Perelli[1], Darío Balan[1], Alfredo Palacios[1], Federico Augustovski[1,2], Andrés Pichón-Riviere[1,2], Andrea Olga Alcaraz[1]

1 Instituto de Efectividad Clínica y Sanitaria (IECS), Buenos Aires, Argentina, 2 Centro de Investigaciones en Epidemiología y Salud Pública (CIESP), Consejo Nacional de Investigaciones Científicas y Técnicas (CONICET), Buenos Aires, Argentina

* abardach@iecs.org.ar

## Abstract

### Background

Approximately two-thirds of Argentine adults are overweight or obese, and 11% have diabetes. Over the last two decades, all population groups have increased their consumption of ultra-processed foods and sugar-sweetened beverages (SSB). We aimed to estimate the disease burden—deaths, events, and costs to the health system—attributed to SSB consumption in Argentina.

### Methods

We used a comparative risk assessment framework to estimate the health and economic impacts that would be avoided in a scenario without sugar-sweetened beverage (SSB) consumption. We calculated the direct effects on diabetes, cardiovascular disease, and BMI, and then estimated the effects of BMI on disease incidence. Finally, we applied the population attributable factor to calculate the health and economic burden avoided in Argentina in 2020.

### Results

Our model estimated that about 4,425 deaths, 110,000 healthy life years lost to premature death and disability, more than 520,000 cases of overweight and obesity in adults, and 774,000 in children and adolescents would be attributed to SSB Consumption in Argentina. This disease burden corresponds to 23% of type-2 diabetes cases and other significant proportions of cardiovascular disease and cancer. The overweight and obesity costs attributable to SSB totaled approximately $47 million in adults and $15 million in children and adolescents.

**Data Availability Statement:** All relevant data are within the paper.

**Funding:** This study received financial support from the International Development Research

Centre (IDRC), Canada, Project Number: 108646-001. The funders had no role in study design, data collection and analysis, decision to publish, or preparation of the manuscript.

**Competing interests:** The authors have declared that no competing interests exist.

## Conclusion

A significant number of disease cases, deaths, and health care costs could be attributed to SSB consumption in Argentina. Implementing measures to reduce the sugar content in beverages is a pending debt for the country and could lead to measurable improvements in population health, especially among children and adolescents.

## Introduction

Approximately 70% of deaths worldwide are caused by non-communicable diseases (NCDs) [1]. The World Health Organization (WHO) estimates that 77% of premature deaths from NCDs occur in low- and middle-income countries [1]. In addition, NCDs are associated with high attributable costs to health systems and society, so these diseases and economic burdens restrict global development to a considerable degree [2]. In 2015 the United Nations defined the Sustainable Development Goals, which regarded prevention and control of non-communicable diseases as core priorities (SDG 3 or SDG 3.4) [3].

Worldwide, eight million deaths in 2019 were attributable to overweight or obesity—two of the most critical determinants of NCD morbidity and mortality [4]. In the past few years, obesity prevalence worldwide has tripled, with over 1.9 billion overweight adults and 650 million obese adults [5]. Childhood obesity and overweight are rising worldwide, particularly in Latin America and the Caribbean [6]. Studies have also shown that overweight and obesity have a high economic cost in developed and developing countries [7, 8]. For instance, Okunogbe et al. reported that by 2019, the economic impact of overweight and obesity ranged from 0.8% of India's total gross domestic product (GDP) to 2.4% in Saudi Arabia [8].

Sugar-sweetened beverages (SSB) are high in calories, low in nutritional value, and are one of the primary sources of added sugars in the diet [9]. Consumption varies considerably by sociodemographic characteristics and is higher among young people, men, and the Americas than in other regions [10, 11]. SSB consumption promotes weight gain and increases the risk of other metabolic disorders, such as type 2 diabetes, in adults [12, 13]. Several studies have shown that reducing SSB consumption reduces the risk of overweight and obesity and thus reduces the risk of obesity-related diseases such as cardiovascular and cerebrovascular disease, cancer, musculoskeletal disease, asthma, depression, social isolation, and dental caries, to name a few [14, 15]. SSB consumption has also been linked to weight gain in children and adolescents, according to WHO [16]. People who are overweight or obese are at greater risk of developing severe health problems, such as type 2 diabetes, high blood pressure, asthma, sleep disorders, liver disease, low self-esteem, and social and emotional issues [17]. Several studies have reported an association between consuming sugar-sweetened beverages and increased incidence and mortality from type 2 diabetes mellitus and cardiovascular disease, regardless of BMI [12, 13, 18]. Sugar raises blood glucose levels and promotes insulin resistance, a risk factor for cardiovascular diseases and type 2 diabetes [12, 13, 18].

Treating all conditions associated with SSB consumption has a significant organizational and economic impact on the health system that needs to be measured.

In the last two decades in Argentina, energy consumption from ultra-processed foods has increased by 53%. It now accounts for more than one-third of the total kcal consumed per day, decreasing the consumption of unprocessed or minimally processed foods [19].

The consumption of SSB is high in Argentina, particularly in children. The results of the Second National Nutrition and Health Survey (ENNyS2), conducted in 2018, showed excess weight is the most severe malnutrition problem, with a prevalence of 13.6% in children under

the age of 5 and 41.1% in the 5–17 age group [20] showing a high consumption of foods of low nutritional quality with high sugar, fat, and salt content, such as sugary drinks, snack products, candies, and bakery products. According to this survey, children and adolescents consume 40% more SSB, twice as much confectionery, and three times as much candy as adults [20].

Furthermore, around 66% of adults suffer from being overweight or obese, and 11% of Argentineans have been diagnosed with diabetes. Also, adults average 85 liters of SSBs per year [21]. Estimates of the health and economic burden of disease of SSB consumption are valuable information inputs for regional policy decision-making, given this high consumption and budgetary constraints of health systems. They facilitate the evaluation of the potential impact of policies such as taxes, front labeling, advertising restrictions, or changes in the school environment [22, 23]. The Pan American Health Organization (PAHO) prioritizes reducing the consumption of SSBs across the region in its regional action plan [24] by recognizing the importance of modifying the food environment, as it influences preferences, purchasing decisions, and eating behaviors. This manuscript presented the results for Argentina obtained through a collaborative project with researchers, policymakers, and scientists from universities, research centers, and public institutions in Argentina (Institute for Clinical Effectiveness and Health Policy), Brazil (ACT Promoção da Saúde and the University of Rio de Janeiro) El Salvador (Ministry of Health), and Trinidad and Tobago (University of the West Indies). This study aimed to estimate the disease burden in terms of deaths, events, Disability-Adjusted Life Years (DALYs), and costs to the health system that may be attributed to SSB consumption in Argentina for 2020.

## Methods

### Model structure

We used a comparative risk assessment framework to estimate the health and economic effects of sugar-sweetened beverage (SSB) consumption in one year. The causal framework of the model forecasts the health effects of SSB consumption through two main pathways. One is mediated by the impact of SSB consumption on body mass index (BMI). The other path reflects the independent effects of SSB consumption on diabetes mellitus type II and cardiovascular disease (see Fig 1). The design of the model and the selection of outcomes followed a systematic review of SSB models available in the international scientific literature [25] and a policy dialogue held in Buenos Aires in 2018, with the participation of 35 experts and decision-makers from seven countries of Latin America and the Caribbean (for more information,

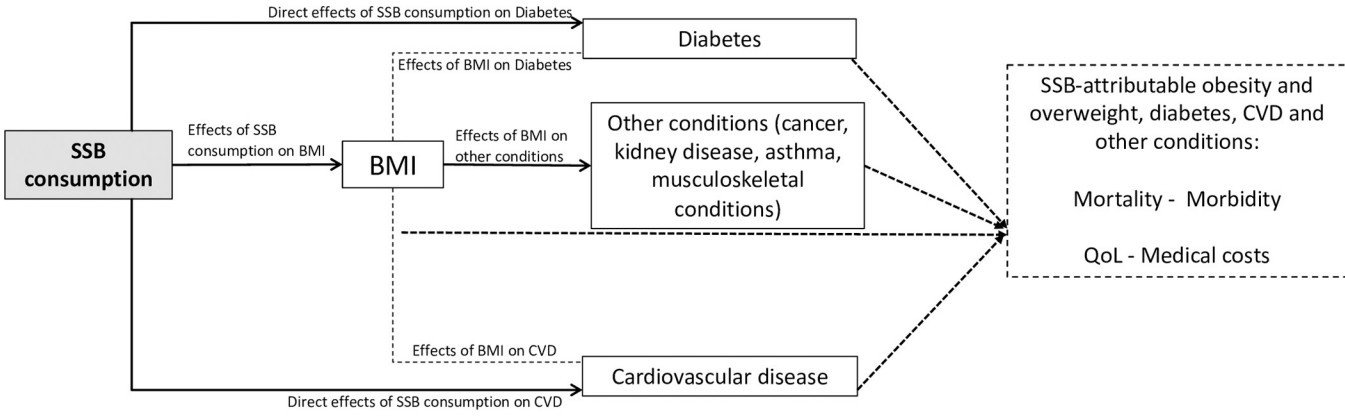

**Fig 1. Model causal framework and main outcomes.**

please visit the official report of this policy dialogue [26]). These activities aimed to shape the development of the model so that it can adequately meet the information needs of decision-makers and contribute to the design and implementation of effective public policies to control SSB in the region.

The model estimated only the impact of SSB consumption on the prevalence of overweight and obesity for children and adolescents without extrapolating the long-term effects on this population.

We used the Population Attributable Fraction (PAF) to calculate the impact of SSB intake on each health event. The PAF estimates the proportional reduction in population disease (incidence or prevalence) or mortality that would occur if exposure to a risk factor (SSB current consumption) were reduced to a minimal theoretical risk exposure scenario (zero consumption). The number of deaths, cases, and costs attributed to a risk factor was quantified by applying the population attributable factor to the total number of deaths, cases, or costs. The equation used to calculate the PAF is as follows (for more information please see S1 File):

$$PAF = 1 - \frac{1}{\sum_{i=1}^{n} RR(Xi; \theta k)}$$

Where $X_i$ represents the consumption of sugar-sweetened beverages (servings per day) by sex and simple age; $\theta k$ corresponds to the relative risk value reported per one unit daily serving increase in sugar-sweetened beverage consumption for each simple age and sex, and finally, RR ($Xi; \theta k$) represents the specific relative risk for each disease by simple age and sex, according to consumption level.

We followed three consecutive steps to estimate the impact of SSB consumption on Argentina's health and economic burden for 2020. As a first step, we calculated the direct effect of changes in SSB consumption on diabetes, CVD, and BMI. Secondly, we estimated the impact of BMI on disease incidence. Finally, we applied the population attributable factor to estimate the health and economic burden attributable to SSB.

The model calculated the number of cases, deaths, DALYs, and costs attributable to SSB. Disease burden was estimated by comparing Argentina's disease events, deaths, and associated costs based on current SSB consumption (baseline) prevalence with a counterfactual scenario without SSB consumption. We used Monte Carlo simulation for the uncertainty analyses. We examined the impact of uncertainty related to SSB consumption and the RR linking SSB with BMI, diabetes, and CVD on the results. The results were generated by 1000 iterations using the distribution of SSB consumption (mean and standard error) and the relative risks for disease incidence (using normal distributions and standard deviations calculated from the 95% confidence intervals) per gender and age group. We reported the mean and 95% confidence interval. We performed our analysis using Stata, version 14.04, and Visual Basic Excel.

## Epidemiological data

Concerning the epidemiological parameters, we prioritized the following sources of data:

1) Argentinean sources when available, 2) Latin American sources when local data was not available, 3) international sources, and 4) estimates from the research group when data was not considered "transferable" from other sources.

**Sugar-sweetened beverage (SSB) consumption.**  We defined SSBs, based on Singh et al. [10], as the following ones: sports and energy drinks, sugar-sweetened sodas, sugar-sweetened fruit juices, and sugar-sweetened and flavored waters. Sugar-sweetened dairy products -like coffee, mate, or tea- and homemade SSBs were not included. One serving was defined as the intake of 240 ml of any SSB.

For SSB consumption, Argentinean official data obtained from national population-based surveys were preferred as sources for children, adolescents, and adults [20, 21, 27]. For children (0–11 years), SSB consumption data was obtained from the second National Nutrition and Health Survey (ENNyS2) [20], while for adolescents (11–18 years), this data was obtained from the Argentinean Global School-based Student Health Survey (2018) [27]. Finally, data on SSB consumption was obtained from the Fourth National Risk Factor Survey for adults [21].

**SSB impact on BMI and other diseases.** We defined overweight and obesity, from BMI values, according to the World Health Organization [28].

An increase of 0.10 (95% CI, 0.05–0.15) $kg/m^2$ in BMI in subjects with basal BMI $<25$, and an increase of 0.23 (95% CI, 0.14–0.32) $kg/m^2$ in BMI in subjects with basal BMI $\geq 25$, was inputted per each serving of SSB consumed per day in adults [29]. Also, BMI reductions of ~0.57 $kg/m^2$ were calculated for a 1.7 servings reduction in SSB in children [30].

Overweight and obesity prevalence data were obtained from the Fourth National Risk Factor Survey (2018) [21]. Data were gathered on each disease's demographic structure, incidence, prevalence, and mortality, stratified by gender and age. We used the DISMOD II software [31] to model missing data regarding incidence, prevalence, or case-fatality rate for each disease, sex, and age when not available. We used different relative risk (RR) values to quantify both the direct path (the association of SSB daily consumption on incidence/mortality of diabetes mellitus type 2 and cardiovascular diseases) and the indirect pathway (effects of SSB consumption on remaining conditions through BMI).

Concerning the direct path, and based on Imamura et al. [12], we assumed a RR value of 1.37 for diabetes type 2 incidence (also assuming that it consequently affects prevalent cases and mortality in the same way) per SSB serving per day. For cardiovascular disease, a RR of 1.08 (95% CI, 1.04 to 1.13) and 1.08 (95% CI, 1.02 to 1.14) per serving per day were considered for incidence/prevalence and mortality, respectively, based on Yin et al. [13].

On the other hand, we modeled the remaining diseases included in the model (esophageal, colon-rectum, uterine body, kidney and gallbladder, and biliary tract cancer, osteoarthritis, low back pain, asthma, dementia, chronic kidney disease, biliary tract, and gallbladder diseases,) through a mediated BMI increase (indirect pathway). Given that a direct RR was already being applied for the association between the consumption of SSBs and cardiovascular events or diabetes events, and these RRs were not adjusted for BMI, we did not include RRs for the association between BMI and cardiovascular events or diabetes to avoid double counting. The respective RR values of developing these diseases from different basal BMI measures were obtained from the Global Burden of Disease study (GBD, Institute for Health Metrics and Evaluation) [4].

Most of the model inputs are differentiated by age and gender: BMI, SSB consumption, population, epidemiological data (incidence, prevalence, mortality), and the RR of BMI on the events (included in the Supporting Information, S1 Table in S1 File). However, no data disaggregated by age and gender was found regarding the association in terms of RR between SSB consumption on BMI, CVD, and diabetes.

## Direct medical costs

The direct medical costs (including diagnosis, treatment, and follow-up) were estimated for each condition considered in the model from a healthcare perspective. These direct medical costs represent a weighted average of the prices of Argentina's three main healthcare subsectors (public, social security, and private). In turn, the costs were differentiated in the first year and subsequent years to distinguish and approximate the difference in the expenses of incident and prevalent cases. The research team estimated the healthcare costs through a micro-costing

approach, depending on the availability and quality of the data. A Microsoft Excel spreadsheet was designed to estimate the direct medical cost of the following health conditions: type 2 diabetes (with and without complications), overweight, obesity type 1, 2, and 3, acute myocardial infarction and its follow-up, heart failure, kidney insufficiency (with and without dialysis) and stroke. These micro-costing spreadsheets were developed based on the clinical experience of local physicians and, when necessary, from local disease management guidelines [32, 33]. The direct medical costs were estimated in the year 2020, in local currency units (Argentinean pesos) and then converted to US dollars using the exchange rates of 2020, ARS 48.14 to 1 US dollar ($), published by the Central Bank of Argentina [34].

## Main epidemiological and economic inputs

The key epidemiological and economic parameters included in the model are shown in Tables 1 and 2. The average SSB consumption in the Argentine population was one serving per day in adults and 1.5 servings per day in children and adolescents, with significant differences between sexes and age groups (see Table 1).

## Results

### Overweight and obesity

Our model estimated that more than 520,000 cases of overweight and obesity in adults and 774,000 in children and adolescents might be attributable to SSB consumption in Argentina. (Table 3). Fig 2 shows the total number of cases of overweight and obesity and the costs attributable to SSB consumption by sex and age group. Fig 2 panel A shows that children and adolescents diagnosed with obesity incurred costs of more than $ 15 million (2020). Boys were more affected than girls concerning the total number of obesity events by 25% (relative terms). Also, boys aged 5 to 11 showed the highest burden of obesity events: almost four times more than the lowest vulnerable group among children (girls aged 0 to 4 group). Men and women aged 18 to 44 years were most affected, accounting for 77% of the total cases among adults (Fig 2, panel B). Also, the overweight and obesity costs attributable to SSB totaled approximately $47 million in adults (2020).

### Cardiovascular, diabetes type 2, and other diseases

Specifically, 23% of all cases of diabetes (639,000 patients in 2020) and other diseases, such as heart disease, cerebrovascular disease, and cancer, with 163,000 affected during this period, could be attributed to SSB consumption in Argentina. As shown in Table 3, type 2 diabetes mellitus would seem to be the disease with the highest burden among diseases attributable to SSB in Argentina, with 19% of the burden of attributable deaths, 22% of DALYs, and 23% of attributable events. Men were more affected than women by 35% concerning events and 32% concerning deaths (relative terms). Also, in terms of costs, approximately $903 million (direct cost of treating diabetes 2) could be saved in the case of non-SSB consumption, representing 23% of the total cost of the disease in Argentina.

Cardiac and cerebrovascular disease account for about 5–6% of attributable costs. Men showed to be more affected in both cases: 50% and 60% more events and deaths for cardiac disease and 40% more events and deaths for cerebrovascular illness than women. Chronic kidney disease, asthma, musculoskeletal disease, and other attributable diseases account for 0.5–1.5% of the burden.

In sum, under a theoretical scenario of zero SSB consumption in Argentina, 4,425 deaths and 110,000 healthy life-years lost to premature death and disability could be averted. The

**Table 1. Key epidemiological parameters.**

| | | | Total Population (millions) | Body Mass Index (Mean) | Mean daily SSB intake (8 oz servings/day) | Overweight (% total population) | Grade 1 Obesity (% total population) | Grade 2 Obesity (% total population) | Grade 3 Obesity (% total population) |
|---|---|---|---|---|---|---|---|---|---|
| Adults | Women | [18–44] | 8.4 | 26.7 | 1.1 | 27.4% | 14.9% | 6.9% | 3.7% |
| | | [45–64] | 4.3 | 29.4 | 0.8 | 29.6% | 27.3% | 9.0% | 6.5% |
| | | [65+] | 2.8 | 29.8 | 0.5 | 33.9% | 26.3% | 10.8% | 6.3% |
| | | Overall | 15.6 | 28.4 | 0.8 | 30.0% | 22.0% | 9.0% | 7.0% |
| | Men | [18–44] | 8.7 | 26.9 | 1.7 | 35.5% | 17.1% | 5.6% | 2.0% |
| | | [45–64] | 4.0 | 29.5 | 1.2 | 42.7% | 27.9% | 9.6% | 2.6% |
| | | [65+] | 2.0 | 29.2 | 0.6 | 43.0% | 28.7% | 9.0% | 2.1% |
| | | Overall | 14.7 | 28.4 | 1.1 | 39.0% | 24.0% | 8.0% | 3.0% |
| | Both | [18–44] | 17.1 | 26.8 | 1.4 | 31.3% | 15.9% | 6.3% | 2.9% |
| | | [45–64] | 8.4 | 29.4 | 1.0 | 35.8% | 27.6% | 9.3% | 4.6% |
| | | [65+] | 4.7 | 29.5 | 0.6 | 38.0% | 27.4% | 10.0% | 4.4% |
| | | Overall | 30.2 | 28.4 | 1.0 | 35.0% | 23.0% | 9.0% | 5.0% |
| Children and adolescents | Girls | [0–4] | 1.8 | NA (*) | 0.7 | 11.0% | 3.1% | | |
| | | [5–17] | 4.5 | | 1.6 | 22.3% | 17.6% | | |
| | | Overall | 6.3 | | 1.4 | 17.0% | 10.0% | | |
| | Boys | [0–4] | 1.9 | | 0.7 | 9.1% | 4.0% | | |
| | | [5–17] | 4.7 | | 1.9 | 19.3% | 22.9% | | |
| | | Overall | 6.7 | | 1.6 | 14.0% | 13.0% | | |
| | Both | [0–4] | 3.8 | | 0.7 | 10.0% | 3.6% | | |
| | | [5–17] | 9.2 | | 1.8 | 20.7% | 20.4% | | |
| | | Overall | 13.0 | | 1.5 | 16.0% | 11.5% | | |
| Sources | | | 1 | 2 | 3 | 2;3 | 2;3 | 2;3 | 2;3 |

Note

(*) This input was not used by the model since the impact of SSB consumption through BMI on other conditions was not modeled in children.

1. Instituto Nacional de Estadística y Censos de la República Argentina. [cited 19 Feb 2021]. Available at: https://www.indec.gob.ar/indec/web/Nivel4-Tema-2-24-84

2. Ministerio de Salud de Argentina. Cuarta Encuesta Nacional de Factores de Riesgo.

Available at: https://bancos.salud.gob.ar/sites/default/files/2020-01/4ta-encuesta-nacional-factores-riesgo_2019_principales-resultados.pdf

3. Local expert estimates, based on the First National Survey of Nutrition and Health, Argentina (ENNYS).

Available at: http://www.extensioncbc.com.ar/wp-content/uploads/ENNyS-2007.pdf)

health system spends $1,153 million each year on the treatment of related diseases, which represents 2.8% of the country's health expenditure. Fig 3 shows the disease burden that could be avoided in a case of non-SSB consumption in adults in Argentina. The most affected group in terms of events are men aged 45 to 64 years. Also, in terms of fatalities, men aged 65 years or more closely followed by women in the same age group.

## Uncertainty analysis

The findings obtained from the uncertainty analysis are presented in Table 3 as 95% CI around the central value of the number of events, deaths, DALYs, and costs for each condition and gender.

**Table 2. Other epidemiological inputs and key economic parameters.**

| Diseases | Incidence rate (per 100,000 population) | Prevalence rate (per 100,000 population) | Mortality rate (per 100,000 population) | Disability weights | Costs per incident event ($) | Costs per prevalent event ($) |
|---|---|---|---|---|---|---|
| Alzheimer disease and dementias | 798 | 4311 | 483 | 0.15 | $ 1,206 | |
| Asthma | 251 | 4118 | 4.84 | 0.04 | $ 869 | |
| Atrial fibrillation and flutter | 139 | 2462 | 63.11 | 0.08 | $ 573 | $ 1,804 |
| Ischemic heart disease | 835 | 3409 | 476 | 0.03 | $ 7,613 | $ 1,209 |
| Chronic kidney disease | 152 | 3019 | 21.15 | 0.04 | $ 869 | |
| Colon and rectum cancer | 101 | 283 | 82.01 | 0.06 | $ 8,780 | $ 1,146 |
| Diabetes Mellitus type 2 | 356 | 12363 | 71.91 | 0.07 | $ 1,414 | |
| Esophageal cancer | 16.39 | 14.20 | 16.16 | 0.20 | $ 13,529 | $ 9,163 |
| Gallbladder and biliary diseases | 245 | 1010 | 12.46 | 0 | $ 162 | |
| Gallbladder and biliary tract cancer | 20.04 | 12.13 | 19.62 | 0.28 | $ 10,952 | $ 7,497 |
| Hypertensive heart disease | 812 | 1296 | 121 | 0.08 | $ 1,058 | |
| Stroke (ischemic and intracerebral hemorrhage) | 184 | 1371 | 133 | 0.15 | $ 1,718 | |
| Subarachnoid hemorrhage | 36.33 | 234 | 17.28 | 0.15 | $ 3,298 | |
| Kidney cancer | 24.40 | 99.26 | 13.92 | 0.07 | $ 11,597 | $ 8,122 |
| Low back pain | 6282 | 15901 | 0 | 0.11 | $ 10 | |
| Osteoarthritis | 346 | 10592 | 0 | 0.03 | $ 223 | |
| Uterine cancer | 21.87 | 120 | 10.41 | 0.07 | $ 5,492 | $ 543 |
| Overweight (< 18 years) | NA | | | | $ 8 | |
| Obesity (< 18 years) | | | | | $ 24 | |
| Overweight (Adults) | | | | | $ 0 | |
| Grade 1 Obesity (Adults) | | | | | $ 10 | |
| Grade 2 Obesity (Adults) | | | | | $ 139 | |
| Grade 3 Obesity (Adults) | | | | | $ 536 | |
| **Country Economic Data** | | | | | | |
| Gross Domestic Product Per Capita (USD) | 9,912 | | | | | |
| Percentage of GDP spending on health | 9.1% | | | | | |
| USD Exchange rate | 48.14 | | | | | |

Source: Argentine Central Bank (2020), own elaboration through micro costing. USD Exchange rate. Argentina: 48.14; Source: http://www.bcra.gov.ar/

For events other than CVD and Cancer the cost represents an average of the annual costs of diagnosis and management for the condition

## Discussion

In this study, through a simulation model tailored for Latin America, we estimated the health and economic burden that could be attributed to SSB consumption in Argentina.

Our model estimated that approximately 4,000 deaths, two million events of illness, 520,000 cases of overweight or obesity in adults, 774,000 of overweight or obesity in children and adolescents, and 1.15 billion US dollars spent in direct medical costs could be attributed to SSB consumption in Argentina.

Our results are consistent with previous studies in both the direction and magnitude of the effects of SSB on health and economic outcomes. The study by Singh et al. [10], which considered only the BMI-mediated effects of SSB consumption, had already found a substantial

**Table 3. Health and economic burden attributed to SSBs consumption in adults: Mortality, disease events, DALYs, and direct medical costs, by cause and sex.**

| Condition | Women | | Men | | Both | |
|---|---|---|---|---|---|---|
| | Total (%*) | CI 95% | Total (%*) | CI 95% | Total (%*) | CI 95% |
| *Obesity and Overweight (<18 y)* | | | | | | |
| Events | 318,855 / 2,053,392 (16%) | [153,592–503,791] | 455,164 / 2,251,656 (20%) | [219,252–719,159] | 774,018 / 4,305,048 (18%) | [372,839–1,223,565] |
| Direct medical cost, USD (million) | $ 5.77 / $29.4 (20%) | [$4.2-$8.2] | $ 9 / $36 (25%) | [$3.6-$15] | $ 14.8 / $ 65.4 (23%) | [$ 8 - $ 23] |
| *Obesity and Overweight (adults)* | | | | | | |
| Events | 169,061 / 9,897,071 (2%) | [117,209–288,925] | 351,702 / 10,212,675 (3%) | [243,835–601,058] | 520,762 / 20,109,746 (3%) | [361,044–890,222] |
| Direct medical cost, USD (million) | $ 28.08 / 619.86 (5%) | [$27-$49] | $ 19.2 / $350 (5%) | [$18-$34] | $ 47.3 / $ 969.8 (5%) | [$ 45.5 - $ 83.2] |
| *Diabetes Mellitus* | | | | | | |
| Events | 281,984 / 1,406,057 (20%) | [126,846–417,421] | 356,887 / 1,349,033 (27%) | [169,944–505,523] | 638,871 / 2,755,090 (23%) | [296,851–922,944] |
| Deaths | 563 / 3,388 (17%) | [246–856] | 754 / 3,633 (21%) | [341–1,114] | 1,317 / 7,021 (19%) | [592–1,970] |
| DALYs | 27,007 / 139,603 (19%) | [12,291–40,445] | 34,925 / 138,834 (25%) | [16,904–50,282] | 61,932 / 278,437 (22%) | [29,901–94,542] |
| Direct medical cost, USD (million) | $ 398.80 / $ 1,988.55 (20%) | [$181 - $597] | $ 504.74 / $1,908 (26%) | [$244-$727] | $ 903 / $ 3,896 (23%) | [$ 420 - $ 1,305] |
| *Cardiac conditions* ** | | | | | | |
| Events | 18,456 / 437,411 (4%) | [8,746–28,439] | 35,119 / 588,503 (6%) | [16,866–53,402] | 53,575 / 1,025,944 (5%) | [25,612–81,840] |
| Deaths | 740 / 18,153 (4%) | [350–1,141] | 1,061 / 18,844 (6%) | [508–1,618] | 1,801 / 36,997 (5%) | [858–2,759] |
| DALYs | 7,225 / 159,545 (5%) | [3,420–11,119] | 15,664 / 208,848 (8%) | [7,528–23,834] | 22,889 / 393,334 (6%) | [9,817–31,540] |
| Direct medical cost, USD (million) | $ 32.6 / $ 772 (4%) | [$15 - $50] | $ 65.1 / $ 1,074 (6%) | [$31-$99] | $ 97.7 / $ 1846 (5%) | [$ 47- $ 149] |
| *Cerebrovascular conditions* | | | | | | |
| Events | 13,606 / 283,518 (5%) | [6,481–20,855] | 15,425 / 231,778 (7%) | [7,453–23,316] | 29,031 / 515,296 (6%) | [13,935–44,171] |
| Deaths | 482 / 11,235 (4%) | [229–743] | 574 / 9,920 (6%) | [275–873] | 1,056 / 21,155 (5%) | [504–1,616] |
| DALYs | 8,072 / 161,099 (5%) | [3,843–12,367] | 10,652 / 153,406 (7%) | [5,155–16,125] | 18,724 / 314,504 (6%) | [9,157–29,284] |
| Direct medical cost, USD (million) | $ 27.90 / $ 572.93 (5%) | [$13-$43] | $29.6 / $435 (7%) | [$15-$45] | $ 57.5 / $ 1,008 (6%) | [$ 28 - $ 87] |
| *Chronic kidney conditions* | | | | | | |
| Events | 11,920 / 1,087,740 (1%) | [7,937–19,198] | 9,508 / 581,952 (2%) | [6,254–15,250] | 21,428 / 1,669,691 (1.3%) | [14,395–33,962] |
| Deaths | 37 / 3,645 (1%) | [25–60] | 53 / 3,536 (2%) | [35–87] | 90 / 7,181 (1.3%) | [60–145] |
| DALYs | 708 / 58,643 (1%) | [457–1,104] | 963 / 52,808 (2%) | [604–1,473] | 1,671 / 111,451 (1.5%) | [1,132–2,723] |
| Direct medical cost, USD (million) | $ 10.37 / $ 945.51 (1%) | [$7-$16] | $ 8.3 / $506 (2%) | [$5-$12] | $ 18.6 / $ 1451.4 (1.3%) | [$ 13 - $ 30] |
| *Asthma* | | | | | | |
| Events | 7,878 / 968,062 (0.8%) | [5,223–12,395] | 8,111 / 626,449 (1%) | [5,330–12,570] | 15,989 / 1,594,510 (1%) | [10,581–25,097] |
| Deaths | 2 / 303 (0.7%) | [2–4] | 2 / 183 (1%) | [1–3] | 4 / 486 (0.9%) | [3–7] |
| DALYs | 334 / 40,245 (0.8%) | [216–513] | 341 / 25,749 (1%) | [220–519] | 675 / 65,994 (1%) | [472–1,116] |
| Direct medical cost, USD (million) | $ 6.86 / $ 841.44 (0.008%) | [$4-$11] | $ 7 / $545 (1%) | [$5-$11] | $ 13.9 / $ 1,386 (1%) | [$ 9- $ 22] |
| *Musculoskeletal Conditions* | | | | | | |
| Events | 14,652 / 5,313,518 (0.3%) | [9,741–23,101] | 19,125 / 4,143,750 (0.5%) | [12,556–29,894] | 33,777 / 9,457,267 (0.4%) | [22,230–53,137] |
| Deaths | 0/0 | [0–0] | 0/0 | [0–0] | 0/0 | [0–0] |
| DALYs | 1,011 / 361,979 (0.3%) | [655–1,554] | 1,386 / 294,635 (0.5%) | [892–2,125] | 2,397 / 656,615 (0.4%) | [2,111–5,045] |

*(Continued)*

**Table 3.** (Continued)

| Condition | Women | | Men | | Both | |
|---|---|---|---|---|---|---|
| Direct medical cost, USD (million) | $ 0.85 / $ 325.86 (0.3%) | [$0.55-$1.31] | $ 0.8 / $195 (0.4%) | [$0,5-$1,2] | $ 1.6 / $ 520.3 (0.3%) | [$ 1 - $ 3] |
| *Other conditions*** | | | | | | |
| Events | 6,311 / 581,030 (1%) | [4,195–9,877] | 2,870 / 280,495 (1%) | [1,889–4,612] | 9,181 / 861,525 (1%) | [6,159–14,421] |
| Deaths | 81 / 19,838 (0.4%) | [53–131] | 76 / 13,122 (0.6%) | [50–128] | 157 / 32,061 (0.5%) | [106–253] |
| DALYs | 910 / 191,118 (0.5%) | [590–1,389] | 1,093 / 155,555 (0.7%) | [694–1,695] | 2,003 / 346,674 (0.6%) | [1,382–3,290] |
| Direct medical cost, USD (million) | $ 3.26 / $ 553.22 (0.6%) | [$2-$5] | $ 3.6 / $489 (0.8%) | [$3-$6] | $ 6.9 / $ 1,042 (0.7%) | [$ 5 - $ 11] |
| *Total* | | | | | | |
| **Events** | 842,722 / 22,027,829 (3%) | [439,970–1,324,2] | 1,253,911 / 20,356,290 (6%) | [726,21–1,924,657] | 2,096,631 / 42,294,119 (5%) | [1,361,844–3,3132,30] |
| **Deaths** | 1,905 / 56,562 (3%) | [1,203–2,654] | 2,520 / 49,238 (5%) | [1,589–3,500] | 4,425 / 105,799 (4.2%) | [2,773–6,146] |
| **DALYs** | 45,267 / 1,112,231 (4%) | [21,472–68,491] | 65,024 / 1,054,778 (6%) | [31,997–96,053] | 110,290 / 2,167,009 (5%) | [53,972–167,540] |
| **Direct medical cost, USD (million)** | $ 508.64 / $ 6,658 (8%) | [$252-$778] | $647 / $5,537 (12%) | [$326-$4,973] | $ 1,152.5 / $ 12,194.6 (10%) | [$ 687.2 - $ 1633.6] |

* Percentages indicate the proportion of all cases of X disease (i.e., diabetes) that could be attributed to SSBs consumption; for example: In Argentina, 513,958 cases of overweight + obesity in adults could be attributable to SSBs consumption, which represents 2.5% of all cases of overweight + obesity in Argentina. The same applied to direct cost results.

** Includes: Atrial fibrillation, Ischemic heart disease, Hypertensive heart disease

*** Includes: Cancer, Alzheimer's, other dementias, Gallbladder, and biliary disease

USD Exchange rate. Argentina: 48.14. Source: https://data.worldbank.org/indicator/PA.NUS.FCRF

absolute and proportional burden of SSB-related mortality and morbidity in LAC compared with other regions in 2010. These findings were interpreted in the context of the low cost of SSB in the area at the time, the suboptimal implementation of taxes and regulation of advertising, and limited access to safe drinking water. Ten years later, conditions are virtually unchanged in many Latin American and Caribbean countries. Also, in line with our findings, the B.A.S.T.A. project ("Bebidas Azucaradas, Salud y Tarifas en Argentina") estimated that over the period 2015–2024, a 10% reduction in SSB consumption would reduce diabetes cases by 13,300 to 27,700 and heart attacks by 2,500 to 5,100. Also, in this study, the most considerable expected reduction in incidence occurred in the youngest group modeled (35–44 years), particularly among men [35].

Countries in Latin America have a high consumption of unhealthy products. This fact is partly responsible for the epidemic of chronic disease that has afflicted the region in recent decades and has been rendered invisible. Latin America and the Caribbean are regions where the burden of illness and death attributable to SSB is disproportionately high. The excessive tendency of sugar consumption in Latin American countries—especially in Argentina—is a traditional fact that may be explained partly by specific population characteristics, such as a greater affinity for sweet taste [36]. There are several natural targets for policy intervention: low prices for SSBs, lack of regulation of advertising, lack of awareness among the population about the risks associated with SSBs, and lack of warning labels on food in many countries, among others.

Argentina recently passed Law 27642 [37] to promote healthy eating, which states that beverages and packaged foods in which the content of critical nutrients and their energy value exceed the established levels must include a black octagonal warning on the main page for each excess nutrient, as follows: "Excess Sugar Content"; "Excess Sodium Content"; "Excess

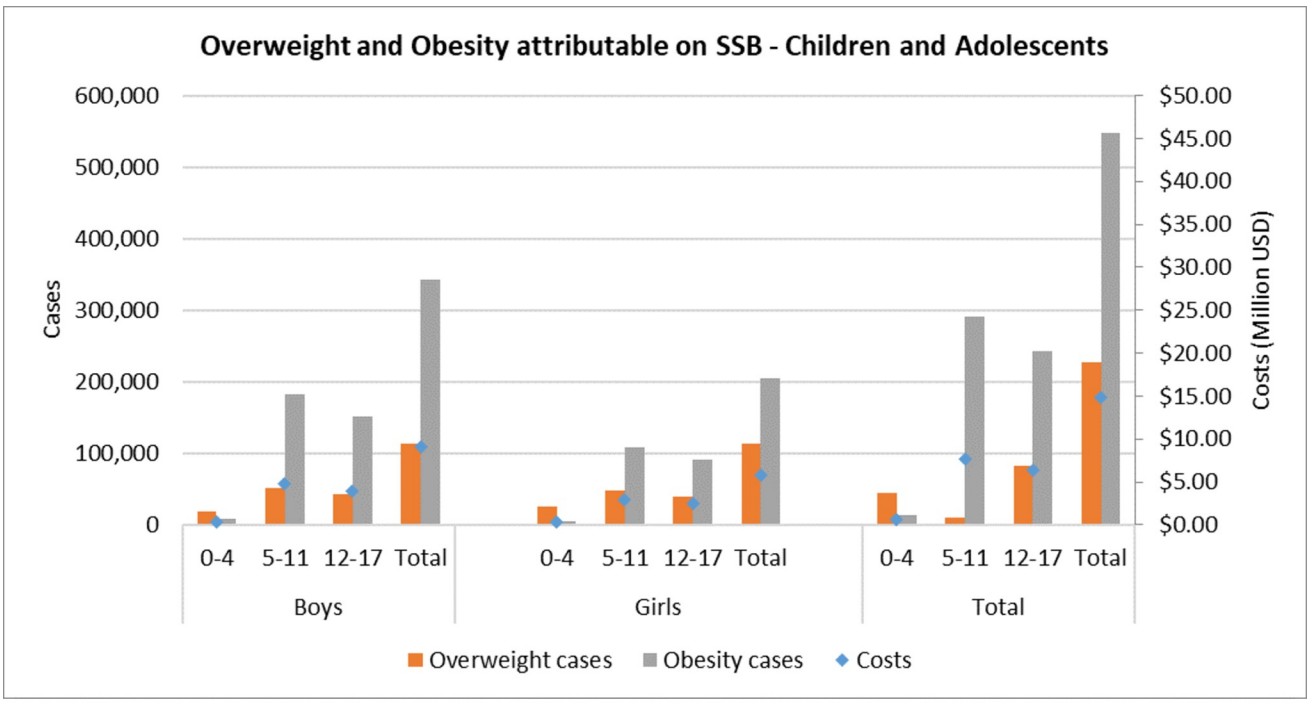

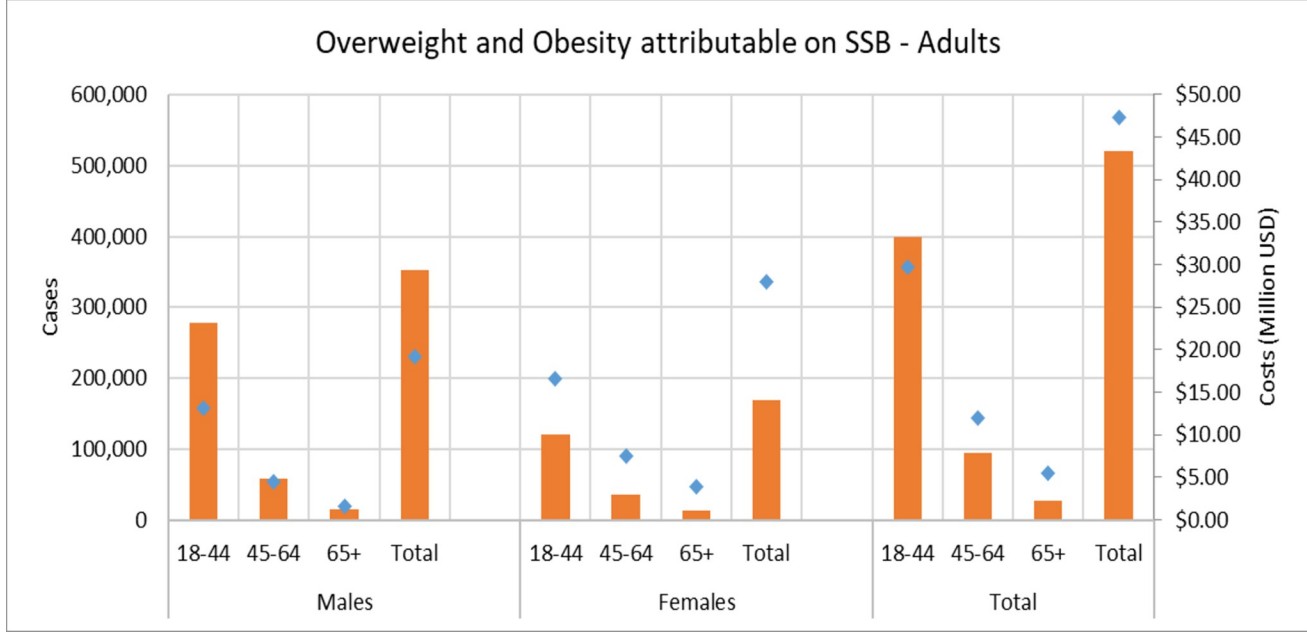

**Fig 2. Overweight and obesity total cases and costs attributable to SSBs consumption by sex and age group.** Panel A: Children and Adolescents. Panel B: Adults. USD Exchange rate. Argentina: 48.14. Source: https://data.worldbank.org/indicator/PA.NUS.FCR.

Saturated Fat Content"; "Excess Total Fat Content"; "Excess Calorie Content." The law also prohibits all forms of advertising, promotion, and sponsorship of food and beverages specifically aimed at children and adolescents and containing at least one warning label. They may not be offered or marketed in educational institutions of the first and second levels of the national education system.

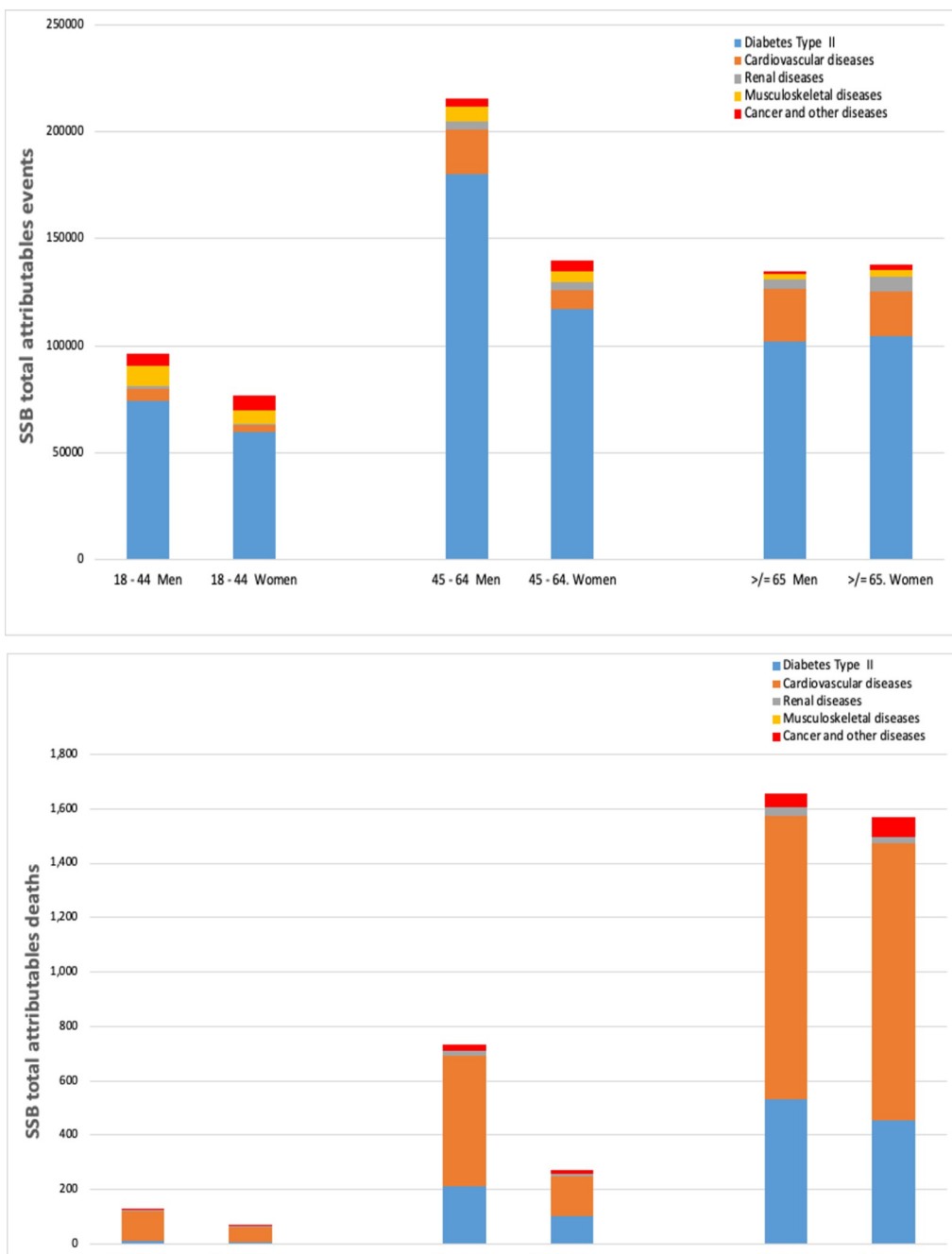

**Fig 3.** Sugar sweetened beverages (SSBs) attributable events (Panel A) and deaths (Panel B) in adults by sex and age.

Although Argentina has recently approved the labeling law, it is essential to start implementing the policy of healthy taxes in Argentina and other countries in the region to decrease consumption [38]. There is evidence that increasing taxes can significantly benefit to the public, and Mexico was the first country to successfully promote this type of change in the region [39].

In Brazil, Claro et al. found that increases in the price of SSBs were associated with a reduction in consumption. As a result of a 1.0% increase in the SSB prices, 0.85% fewer calories were consumed [40].

Furthermore, reformulating SSBs to reduce sugar content is a debt for several countries and could result in measurable reductions in consumption in Latin America [39].

There is already evidence of the effectiveness of labeling, which would be a measure that could be applied to sugar-sweetened drinks. For example, research has been conducted for Mexico on the positive public health impact that warning labeling could have in effectively reducing obesity and its associated costs. This used data from national consumption surveys and the expected effects of labeling, based on experimental studies showing a reduced calorie consumption of up to 10.5% in the case of beverage labeling. Mexico, Chile, Peru, and Uruguay have already introduced labeling for highly processed foods and are starting to benefit from these changes [41].

These and other studies that address the burden of disease can contribute to drawing the attention of the public and policymakers to this crucial public health problem and serve as an argument to support the policies that many countries strive to implement. These burden of disease studies, in turn, are a necessary step for the developing studies that can measure the expected impact of various interventions.

A quasi-experimental study comparing the consumption of sugar-sweetened beverages with the trend before the passage of Chile's pioneering law on front-end labeling and advertising restrictions on labeled products found a reduction of 22.8 ml/day and 11.9 calories/day in the consumption of beverages with excess sugars, with a more marked relative reduction at higher socioeconomic levels [42]. One of the most critical aspects of this research was the enormous impact on the decrease in consumption of beverages with at least one warning label (-23.7%) of this measure compared to the tax increases that had previously occurred in Chile (-3.4%) and Mexico (-7.6%) [42]. A similar law has recently been passed and regulated in Argentina, although its regulations have not yet come into force.

Our investigation had several strengths. We developed a model adapted to the availability and quality of epidemiological and cost data in Argentina and formally addressed the information needs of policymakers in the region. We included sound evidence on the direct effects of SSB on BMI, diabetes mellitus, cardiovascular disease, and BMI on overweight and obesity-related conditions. The inclusion of the direct impact of SSB on cardiovascular disease in our estimates is an innovation over the most previous burden of disease studies. It has been instrumental in producing results that more accurately reflect the epidemiological reality. One of the main limitations of our study is that the quality and availability of some epidemiological parameters for Argentina are not optimal. However, our results show that SSBs are responsible for a substantial disease burden, probably much higher than our estimates. For instance, in our analysis, we only considered the main health consequences of SSB. Still, other vital dimensions, such as the impact on oral cavities or the social implications for children and adults due to obesity, could play an important role. Moreover, we only consider the direct medical costs spent that could be avoided. These are only a part of the total financial burden when other dimensions such as lost productivity are considered. As another limitation, our study estimated the attributable burden of disease considering a hypothetical counterfactual scenario of zero SSB consumption. It could be possible that some of the burden of disease that we reported as attributed to SSB consumption still exists if people switch from SSB consumption to other harmful food or drink (due to a probable caloric compensation) in a scenario of no SSB consumption. However, this does not mean that the current disease burden attributable to the consumption of SSB is less than that reported in our analysis.

Research gaps include analyzing the direct/indirect costs of SSB exposure in Latin American countries, determining the expected impact on obesity of enforcing formal bans on SSB advertising in the media and social networks, assessing the impact of labeling in Argentina, and examining the combined effect of different food policies enacted together.

## Conclusion

The present study results show that SSB consumption causes a significant number of diseases, deaths, and health care costs in Argentina. This study is the first to assess the financial burden of SSB consumption in the country, including children and a wide range of attributable conditions.

## Supporting information

**S1 File.**
(DOCX)

## Acknowledgments

The authors thank Mr. Daniel Comandé, librarian at IECS for their help with the search strategies, Dr Agustín Ciapponi, for his help in the initial phases of the study, BSc Natalia Elorriaga for her support with public health nutrition, and Bsc Mariana Comolli for her help with dissemination of findings.

## Author Contributions

**Conceptualization:** Ariel Esteban Bardach, Alfredo Palacios, Federico Augustovski, Andrés Pichón-Riviere, Andrea Olga Alcaraz.

**Data curation:** Federico Rodríguez Cairoli, Darío Balan, Alfredo Palacios, Andrés Pichón-Riviere.

**Formal analysis:** Natalia Espínola, Andrés Pichón-Riviere, Andrea Olga Alcaraz.

**Funding acquisition:** Andrés Pichón-Riviere, Andrea Olga Alcaraz.

**Investigation:** Lucas Perelli, Federico Augustovski, Andrés Pichón-Riviere, Andrea Olga Alcaraz.

**Methodology:** Ariel Esteban Bardach, Natalia Espínola, Federico Rodríguez Cairoli, Lucas Perelli, Darío Balan, Alfredo Palacios, Federico Augustovski, Andrés Pichón-Riviere, Andrea Olga Alcaraz.

**Project administration:** Ariel Esteban Bardach, Andrea Olga Alcaraz.

**Resources:** Andrea Olga Alcaraz.

**Supervision:** Ariel Esteban Bardach, Andrés Pichón-Riviere, Andrea Olga Alcaraz.

**Writing – original draft:** Ariel Esteban Bardach, Natalia Espínola, Federico Rodríguez Cairoli, Lucas Perelli, Federico Augustovski, Andrés Pichón-Riviere, Andrea Olga Alcaraz.

**Writing – review & editing:** Ariel Esteban Bardach, Natalia Espínola, Federico Rodríguez Cairoli, Lucas Perelli, Darío Balan, Alfredo Palacios, Federico Augustovski, Andrés Pichón-Riviere, Andrea Olga Alcaraz.

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
