## [Decision Letter · Decision Letter 0]

5 Apr 2022

PONE-D-22-04375The Burden of Disease and Economic Impact of Sugar-Sweetened Beverages’ Consumption in Argentina: a Modeling StudyPLOS ONE

Dear Dr. Bardach,

Thank you for submitting your manuscript to PLOS ONE. After careful consideration, we feel that it has merit but does not fully meet PLOS ONE’s publication criteria as it currently stands. Therefore, we invite you to submit a revised version of the manuscript that addresses the points raised during the review process.

We look forward to receiving your revised manuscript.

Kind regards,

Vijay S. Gc, PhD

Academic Editor

PLOS ONE

Journal Requirements:

Reviewers' comments:

Reviewer's Responses to Questions

**Comments to the Author**

1. Is the manuscript technically sound, and do the data support the conclusions?

Reviewer #1: Partly

Reviewer #2: Yes

2. Has the statistical analysis been performed appropriately and rigorously? 

Reviewer #1: I Don't Know

Reviewer #2: Yes

3. Have the authors made all data underlying the findings in their manuscript fully available?

Reviewer #1: No

Reviewer #2: Yes

4. Is the manuscript presented in an intelligible fashion and written in standard English?

Reviewer #1: No

Reviewer #2: No

5. Review Comments to the Author

Reviewer #1: This paper is a modelled study of the health and economic impact of SSB consumption in Argentina. There has clearly been much work into the development and execution of the model and the graphs and tables are great. With significant revision (minor and some major) and refinement of the use of the English language throughout, this paper has the potential to provide a good addition to the literature.

It is difficult to provide all details to such a complex modelling exercise however, a robust scientific paper is one that other research teams can obtain the same results through the same data sources, software, stakeholders, literature, reports etc. It is always useful to keep this in mind when doing scientific writing. Some specific considerations:

• Abstract:

Methods: The two last sentences should be in past tense. Please revise.

Results: It might be useful to say in the first instance of the results that ‘our model estimated that because of SSB consumption…..’.

• Introduction:

First paragraph, first sentence: needs citing.

First paragraph, last sentence: would be good to include which SDG regarded prevention and control of NCDs (i.e. SDG 3 or 3.4).

Second paragraph, first sentence, change the tense. So….’Worldwide, eight million deaths were attributed to overweight or obesity in 2019…… The sentence reads poorly. Please revise. Eg: Worldwide, eight million deaths in 2019 were attributable to overweight or obesity – two of the most critical determinants of NCD morbidity and mortality.’

Second paragraph, second sentence: what are the gender differences? This is too general. Either give values or delete. Similarly, what are some of the economic costs you speak of? Some stats would be useful for the reader.

Third paragraph, second sentence needs citing. There appears to be a lot of duplication in information in this paragraph. Suggest revising and citing correctly throughout.

Fourth paragraph: Results from studies done in the past should be written in past tense. Please revise.

Fourth, fifth and sixth paragraphs need revising. For instance, the authors begin paragraph seven by saying children and adolescents consume 40% more SSBs, confectionery and then go on to give stats for adult obesity. Would be much better to give stats at this point for child and adolescent obesity instead of talking about it in the fourth paragraph.

Seventh paragraph: You need to spell out PAHO in full the first time you use it and then use acronyms. Also, how have PAHO prioritized the need to reduce SSB consumption? It’s a very general statement and doesn’t really tell the reader anything, as it is currently is. Suggest further work.

• Methods

The authors describe in this section the two causal pathways in the model (ie. BMI and SSB consumption). However, there is no mention of how (or what) determines change in SSB consumption or how it directly effects overweight and obesity independently of BMI and this is an important and relatively new concept to understand. Some (cited) biochemical/pathological background from the global literature on this concept would be useful in the introduction to highlight just how damaging SSBs are. It would help build the argument of why the modelling is important. What determined the change in consumption? Literature? It is unclear what the difference is between the baseline and counterfactual and why that has been chosen for the model. Some explanation of this would make the results much clearer for the reader.

Figure 1 is a good figure. Describing the other diseases is useful for the reader.

First paragraph, fifth sentence needs more detail on the conceptualisation of the model. Did the authors have the policy diaglogue? Attend the dialogue? Or is the policy dialogue a report? Who hosted the dialogue? What was the dialogue on? How did these ‘activities’ help shape the model’s development? In what way? Did you follow a protocol? Did you have aims and objectives? What were the needs of the decision-makers? What policies are the ones you speak of to control SSB consumption?

For the PIFs, does it estimate the proportional reduction in incidence or prevalence of disease or mortality? The second sentence in this paragraph needs revising.

Throughout the methods section: your study was done in the past so you need to write it in the past tense (e.g., ….secondly, we estimated the effect of BMI on disease incidence…..Finally we applied….. The model calculated…

Fourth paragraph: should read ‘the estimation of the impact of SSB consumption….. please revise.

What other sensitivity analyses did you estimate? Is the STATA code available for your analyses?

The authors indicate results were generated by 1000 iterations using the distribution of SSB consumption and RRs for each country. Is the model estimating outputs for more than one country or is this an error?

• Epidemiological data

Please provide details of the data sources. What were the surveys? What were the international sources? Where did the authors obtain SSB consumption data from? What is the ‘latest nation risk factor survey’ actually called? Did the authors do any ‘fiddling’ within DISMOD? Could an independent person repeat this method and get the same results? How did you ‘identify’ the RR values? Why did you choose the diseases you did and not others?

The section could be written in a much clearer way. Suggest revising. For instance, the last sentence about the RRs from GBD (which needs to be spelt out in full the first time it’s used) should go before you begin talking about ‘identifying RRs’, the readers then know where the authors obtained the RRs from and makes it clear that is how they ‘identified’ them (if indeed it was where the values for T2DM and CVD were obtained from, it’s really not clear).

• Direct medical costs

The authors speak of macro- and micro-costing approaches but only describe the micro-costing approach. What is the macro-costing approach? Where do these concepts originate? Are they backed by theory? Previous modelling studies? Please provide more detail.

Fifth sentence: please either name the previous cost estimation projects, clinical guidelines and local experts or provide a reference for each for the reader.

• Results

Overall, the authors need to describe their findings in much more detail. It is clear they have gone to the trouble to make detailed tables, but the text descriptions to highlight the significant findings for each would improve this section a lot. Compare and contrast. For instance, the authors indicate boys were more affected than girls. By how much? How much higher was the 5-11 age group’s burden compared to the lowest? How much more affected were adults 18-44 compared to older groups? 47 million. Is that USD?

Table 1

Please provide a definition for the reader of the different grades of obesity, either in the text or in the table or as a table footnote. Different countries have different thresholds for obesity and it is important to provide explicit detail for all tables and their content. Nowhere in the paper have the authors given any definition of the different BMI groups and what constitutes underweight, overweight or anything about the different ‘grades’ of obesity. Please revise for those readers not familiar with BMI ranges. Similarly, the authors may like to explain the reason for not having BMI means for children? Doing so will give this table a lot more meaning and relevance.

Table 2.

Please check the rounding of values for incidence, prevalence and mortality. It may be preferable to have rates per 100,000 as whole numbers when above 100. Similarly, for very low numbers (e.g., 0.0) it may be better to have increased decimal points.

Please ensure units for numbers are included in tables. Overweight and obesity have <18 in the ‘diseases’ column, but what is that? Age? BMI? It is unknown without units.

Are the costs First Year (Acute Event) per capita? If so, please make clear.

Table 3.

Please check your units. Some dollars have no $ signs. Also, what do the two values for the direct medical cost, USD (million) represent? Is one Argentinian dollars and one USD? If so, please make this very clear for the reader within the table or as a footnote.

Or is it per capita/ for the whole population? As it is, no one knows.

Be careful with the text. First sentence after Table 3 shows – 639,000 which makes it look like a negative number but it’s just punctuation. Could you use brackets instead or use the dash properly.

Could you add up T2DM and CVD costs etc to show even more significant results and the significance of multiple morbidities?

Figures 2 & 3

These are nice graphs, particularly Figure 3. Where did you source the data from? This should be included in your paper.

• Discussion

Overall, the discussion needs revising to improve the flow of information. There are some interesting points the authors make, but it’s adhoc and disjointed.

Paragraph 1, use past tense for studies already done. So, ‘In this study, we estimated….’

Paragraph 3: What is BASTA? Please spell out acronyms if only being used once or twice.

Paragraphs 4 & 5: Why are the authors discussing the strengths and limitations in the middle of your discussion? These go at the end of your discussion.

Paragraph 6: Why do countries in Latin America have a high consumption of unhealthy products?

Paragraphs 7 & 8: These would be better in the introduction.

The paper has some interesting, significant findings. With revision, it will be much improved and useful for fellow modellers. All the best with the revision.

Reviewer #2: Thank you for the opportunity to review this paper, entitled ‘The Burden of Disease and Economic Impact of Sugar-Sweetened Beverages Consumption in Argentina: a Modeling Study’. The authors estimate the burden of disease and costs imparted by SSB consumption in Argentina. The authors have used a strong modelling approach to estimate this, and the both the rationale and modelling techniques appear to be sound. However, much of the manuscript should be restructured to increase comprehension and readability. Further, the authors may wish to consider their interpretation of the model, as rather than estimating the exact burden of SSBs, it instead estimates the reduce burden if consumption were reduced to a predetermined level. Finally, should this paper be accepted for publication, I suggest further editing for English language and readability.

Abstract

- As with much of the results and the discussion, please reframe to highlight that the model does not specifically estimate the cases of disease and deaths caused by SSB consumption (for example, of the 4,425 deaths mentioned, in the real world they are not only caused by SSB consumption but reducing SSB consumption may prevent them).

- Reformulating SSBs is not a good policy option to highlight in the abstract. I suggest discussing SSB taxation, marketing restrictions, or food and nutrition policies broadly.

Introduction

- The introduction is too long, and the structure makes it difficult to identify the issue being discussed. I suggest restructuring the introduction into three paragraphs:

1) Overweight, obesity and NCDs are a serious health concern globally.

2) An unhealthy diet, and consumption of SSBs is a key driver of overweight, obesity and NCDs. These are increasing worldwide.

3) In Argentina, consumption of SSBs is both high and increasing, and this is leading to an increased burden of disease.

4) Study rationale and aims

- Discussion of current policies to address unhealthy diets in Argentina should be left for the discussion.

Methods

- I feel that adding additional subheadings to the methods might clarify the approach for an audience with less economic modelling experience. For example, different sections for each of the three stages of estimating health and economic outcomes.

- Instead of a sperate section on data sources, please describe each data source as its use is first mentioned in the text.

- Please define DALYs when first used

Results

- The main epidemiological and economic parameters included in the model should be presented in the methods, not the results (assuming that these are derived from pre-existing data sets).

- Please edit the results to use people-first language. For example, instead of “Overweight and obesity burden children and adolescents the most”, use “Children and adolescents face the greatest burden of overweight and obesity”.

- Similarly, cases of obesity are not attributed to children/adolescents – children/adolescents are living with overweight and obesity.

- In table 3, please present ‘events’ before ‘deaths’

- It is difficult to say that SSBs cause 23% of all cases on diabetes, as so many factors contribute to disease. Instead, what the model estimates is that, if no SSBs were consumed, 23% of diabetes cases would be prevented. Please edit the results throughout to reflect this.

- I suggest not comparing the burden of disease across different NCDs as this implies that some are more important than others. Simply say how many cases of each would be avoided if SSB consumption was reduced to 0.

- Please reference the results of the included sensitivity analyses

Discussion

- As with the results, please be careful about directly attributing costs/deaths to SSB consumption – the model simply estimates that if SSB consumption were reduced, a certain number of costs and deaths would be prevented.

- Please mode the strengths and limitations to later in the discussion (just before the conclusion).

- The authors state (in the limitations section) that their estimates are likely an underestimation, but it is not clear why this is the case – is this because oral health was not included? Please explain further.

- Overall, the discussion would be improved by additional sections on drivers of SSB consumption and further consideration of policies. The authors discuss many policy options for addressing SSB consumption in Argentina, they may want to condense this a little and instead include more discussion on a) why SSB consumption is high (there is some discussion of advertising and prices, but there should be more on this) and b) how to implement best-practice policies (notably how to limit SSB industry influence).

6. PLOS authors have the option to publish the peer review history of their article (what does this mean?). If published, this will include your full peer review and any attached files.

Reviewer #1: No

Reviewer #2: No

---

## [Author Response · Author response to Decision Letter 0]

17 Jun 2022

Reviewer #1: This paper is a modelled study of the health and economic impact of SSB consumption in Argentina. There has clearly been much work into the development and execution of the model and the graphs and tables are great. With significant revision (minor and some major) and refinement of the use of the English language throughout, this paper has the potential to provide a good addition to the literature.

It is difficult to provide all details to such a complex modelling exercise however, a robust scientific paper is one that other research teams can obtain the same results through the same data sources, software, stakeholders, literature, reports etc. It is always useful to keep this in mind when doing scientific writing. Some specific considerations:

Thank you for these comments. We agree on the importance of the transparency and reproducibility of findings in research. We have edited the manuscript and refined the English language use, following your suggestions. 

• Abstract:

Methods: The two last sentences should be in past tense. Please revise.

Thank you. We have already corrected these typos.

Results: It might be useful to say in the first instance of the results that ‘our model estimated that because of SSB consumption…..’.

Thank you. We totally agree with this suggestion and we have already corrected it.

• Introduction:

First paragraph, first sentence: needs citing.

Thanks. It is done now.

First paragraph, last sentence: would be good to include which SDG regarded prevention and control of NCDs (i.e. SDG 3 or 3.4).

Thanks. It is done now.

Second paragraph, first sentence, change the tense. So….’Worldwide, eight million deaths were attributed to overweight or obesity in 2019…… The sentence reads poorly. Please revise. Eg: Worldwide, eight million deaths in 2019 were attributable to overweight or obesity – two of the most critical determinants of NCD morbidity and mortality.’

Thanks. We have edited the sentence after your suggestion.

Second paragraph, second sentence: what are the gender differences? This is too general. Either give values or delete. Similarly, what are some of the economic costs you speak of? Some stats would be useful for the reader.

Thanks. To avoid excess in the number of words, we decided to omit the statement related to gender differences in the introduction. Also, we are now providing some statistics related to economic costs.

 Third paragraph, second sentence needs citing. There appears to be a lot of duplication in information in this paragraph. Suggest revising and citing correctly throughout.

 Thanks. We have cited these statements correctly now.

Fourth paragraph: Results from studies done in the past should be written in past tense. Please revise.

Thanks. We have edited it.

Fourth, fifth and sixth paragraphs need revising. For instance, the authors begin paragraph seven by saying children and adolescents consume 40% more SSBs, confectionery and then go on to give stats for adult obesity. Would be much better to give stats at this point for child and adolescent obesity instead of talking about it in the fourth paragraph.

Thanks. We relocated ENNyS2 data

Seventh paragraph: You need to spell out PAHO in full the first time you use it and then use acronyms. 

Thanks. Done

Also, how have PAHO prioritized the need to reduce SSB consumption? It’s a very general statement and doesn’t really tell the reader anything, as it currently is. Suggest further work.

Thanks. We added some strategies highlighted in the PAHO document

• Methods

The authors describe in this section the two causal pathways in the model (ie. BMI and SSB consumption). However, there is no mention of how (or what) determines change in SSB consumption or how it directly effects overweight and obesity independently of BMI and this is an important and relatively new concept to understand. Some (cited) biochemical/pathological background from the global literature on this concept would be useful in the introduction to highlight just how damaging SSBs are. It would help build the argument of why modeling is important. What determined the change in consumption? Literature? It is unclear what the difference is between the baseline and counterfactual and why that has been chosen for the model. Some explanation of this would make the results much clearer for the reader.

Thanks for your comments. 

- We now added a paragraph in the introduction explaining the effects of consumption of SSB on BMI and SSB on diabetes II and CVD independent of BMI. 

- The model estimates a hypothetical change in the consumption of SSB. The baseline is the current consumption of SSB in Argentina, and the counterfactual is the zero consumption. This statement is now explained better in the estimation of the PIF: “The PIF estimates the proportional reduction in population disease (incidence or prevalence) or mortality would occur if exposure to a risk factor (SSB current consumption) were reduced to a minimal theoretical risk exposure scenario (zero consumption).”

Figure 1 is a good figure. Describing the other diseases is useful for the reader.

Thanks. We appreciate your comment.

First paragraph, fifth sentence needs more detail on the conceptualisation of the model. Did the authors have the policy diaglogue? Attend the dialogue? Or is the policy dialogue a report? Who hosted the dialogue? What was the dialogue on? How did these ‘activities’ help shape the model’s development? In what way? Did you follow a protocol? Did you have aims and objectives? What were the needs of the decision-makers? What policies are the ones you speak of to control SSB consumption?

Thank you for your comment. We have now cited the report of the policy dialogue carried out in the framework of this project where the answers to these questions can be found. (available on: https://www.iecs.org.ar/wp-content/uploads/Informe-Policy-dialogue-Azucar.pdf)

For the PIFs, does it estimate the proportional reduction in incidence or prevalence of disease or mortality? The second sentence in this paragraph needs revising.

Thanks for your comment. We are explaining better now this statement: 

 “The PIF estimates the proportional reduction in population disease (incidence or prevalence) or mortality would occur if exposure to a risk factor (SSB current consumption) were reduced to a minimal theoretical risk exposure scenario (zero consumption).”

Throughout the methods section: your study was done in the past so you need to write it in the past tense (e.g., ….secondly, we estimated the effect of BMI on disease incidence…..Finally we applied….. The model calculated…

Thanks. We have corrected it.

Fourth paragraph: should read ‘the estimation of the impact of SSB consumption….. please revise.

Thanks. We have corrected it.

What other sensitivity analyses did you estimate? Is the STATA code available for your analyses?

Thanks. We did not carry out multiple sensitivity analyses. Now we are stating: “We used Monte Carlo simulations for the sensitivity analysis” (instead of reporting “sensitivity analyses”).

As mentioned in the manuscript, the sensitivity analysis was programmed in STATA and Visual Basic. Programming codes are available upon request by the reader.

The authors indicate results were generated by 1000 iterations using the distribution of SSB consumption and RRs for each country. Is the model estimating outputs for more than one country or is this an error?

Yes, this was an error. We have now corrected it. Thanks for your comment. 

• Epidemiological data

Please provide details of the data sources. What were the surveys? What were the international sources? Where did the authors obtain SSB consumption data from? What is the ‘latest nation risk factor survey’ actually called? 

Thanks. We have edited this paragraph according to your suggestions.

Did the authors do any ‘fiddling’ within DISMOD? 

No. DISMOD was mainly used to calculate case fatality rate values from incidence and mortality data of the included diseases. 

Could an independent person repeat this method and get the same results? How did you ‘identify’ the RR values? Why did you choose the diseases you did and not others?

The section could be written in a much clearer way. Suggest revising. For instance, the last sentence about the RRs from GBD (which needs to be spelt out in full the first time it’s used) should go before you begin talking about ‘identifying RRs’, the readers then know where the authors obtained the RRs from and makes it clear that is how they ‘identified’ them (if indeed it was where the values for T2DM and CVD were obtained from, it’s really not clear).

Thanks for this comment. We have edited this paragraph according to your suggestions.

• Direct medical costs

The authors speak of macro- and micro-costing approaches but only describe the micro-costing approach. What is the macro-costing approach? Where do these concepts originate? Are they backed by theory? Previous modeling studies? Please provide more detail.

Thank you for your comment. This project was carried out not only in Argentina but also in three other Latin American countries. For the Argentina case (purpose of this manuscript), we only used a micro-costing approach. We made a mistake when writing this paragraph, but now we only mention the micro-costing method.

Fifth sentence: please either name the previous cost estimation projects, clinical guidelines and local experts or provide a reference for each for the reader.

We appreciate your comment. In line with our response to the previous comment, we agree we can be more specific about what was done specifically for the case of Argentina. All micro-costings were carried out between a research economist and at least one local medical expert for each disease. Local medical experts suggested the annual resources that these patients usually need. This information was based on their daily experience and, in some cases, on local practice guidelines (diabetes and obesity). Therefore we have edited this paragraph and added the citations for these two local guidelines.

• Results

Overall, the authors need to describe their findings in much more detail. It is clear they have gone to the trouble to make detailed tables, but the text descriptions to highlight the significant findings for each would improve this section a lot. Compare and contrast. For instance, the authors indicate boys were more affected than girls. By how much? How much higher was the 5-11 age group’s burden compared to the lowest? How much more affected were adults 18-44 compared to older groups? 47 million. Is that USD?

Thank you for your contribution. We have already added this information.

Table 1

Please provide a definition for the reader of the different grades of obesity, either in the text or in the table or as a table footnote. Different countries have different thresholds for obesity and it is important to provide explicit detail for all tables and their content. Nowhere in the paper have the authors given any definition of the different BMI groups and what constitutes underweight, overweight or anything about the different ‘grades’ of obesity. Please revise for those readers not familiar with BMI ranges. Similarly, the authors may like to explain the reason for not having BMI means for children? Doing so will give this table a lot more meaning and relevance.

Thanks. We have now explained that the division by BMI values is the same as the World Health Organization. Regarding the mean point BMI in children, this input was not used by the model since the impact of SSB consumption through BMI on other diseases was not modeled in children.

Table 2.

Please check the rounding of values for incidence, prevalence and mortality. It may be preferable to have rates per 100,000 as whole numbers when above 100. Similarly, for very low numbers (e.g., 0.0) it may be better to have increased decimal points.

Please ensure units for numbers are included in tables. Overweight and obesity have <18 in the ‘diseases’ column, but what is that? Age? BMI? It is unknown without units.

Are the costs First Year (Acute Event) per capita? If so, please make clear.

Thanks. We have edited this table according to your suggestions.

Table 3.

Please check your units. Some dollars have no $ signs. Also, what do the two values for the direct medical cost, USD (million) represent? Is one Argentinian dollars and one USD? If so, please make this very clear for the reader within the table or as a footnote.

Or is it per capita/ for the whole population? As it is, no one knows.

Thanks. We now explain in the footer of the table that we first present the value attributed to the consumption of sugar-sweetened beverages for both health outcomes and direct costs. We also offer the percentage that this value represents of the total number of events or deaths or direct costs that this disease brings to Argentina (not only what is attributed to sugary drinks). 

In this case, the total direct cost in the denominator represents all the expenditures in Argentina for this disease.

Be careful with the text. First sentence after Table 3 shows – 639,000 which makes it look like a negative number but it’s just punctuation. Could you use brackets instead or use the dash properly. Could you add up T2DM and CVD costs etc to show even more significant results and the significance of multiple morbidities?

Thanks. It was edited as you suggest.

Figures 2 & 3

These are nice graphs, particularly Figure 3. Where did you source the data from? This should be included in your paper.

Thanks. We appreciate your comment. Figure 2 and 3 show findings estimated through the model.

• Discussion

Overall, the discussion needs revising to improve the flow of information. There are some interesting points the authors make, but it’s adhoc and disjointed.

Paragraph 1, use past tense for studies already done. So, ‘In this study, we estimated….’

Thanks. It is done.

Paragraph 3: What is BASTA? Please spell out acronyms if only being used once or twice.

Thanks. It is the acronyms of a previous model published in the field in Argentina (B.A.S.T.A., for “Bebidas Azucaradas, Salud y Tarifas en Argentina”. We have now corrected it.

Paragraphs 4 & 5: Why are the authors discussing the strengths and limitations in the middle of your discussion? These go at the end of your discussion.

Thanks. It is done.

Paragraph 6: Why do countries in Latin America have a high consumption of unhealthy products?

Thanks. It is done.

Paragraphs 7 & 8: These would be better in the introduction.

We consider that what is related to the policies in this field in Argentina and the region can be better addressed in the discussion section. Also, the second reviewer suggested not addressing the issue of implemented or unimplemented policies in the introduction section.

The paper has some interesting, significant findings. With revision, it will be much improved and useful for fellow modelers. All the best with the revision.

Thanks. We appreciate your comments. We totally agree that your suggestions help a lot in improving this manuscript.

Reviewer #2: Thank you for the opportunity to review this paper, entitled ‘The Burden of Disease and Economic Impact of Sugar-Sweetened Beverages Consumption in Argentina: a Modeling Study’. The authors estimate the burden of disease and costs imparted by SSB consumption in Argentina. The authors have used a strong modelling approach to estimate this, and both the rationale and modelling techniques appear to be sound. However, much of the manuscript should be restructured to increase comprehension and readability. Further, the authors may wish to consider their interpretation of the model, as rather than estimating the exact burden of SSBs, it instead estimates the reduced burden if consumption were reduced to a predetermined level. Finally, should this paper be accepted for publication, I suggest further editing for English language and readability.

Thanks. We appreciate your comments. We have worked on your revisions and also in editing English language and readability.

Abstract

- As with much of the results and the discussion, please reframe to highlight that the model does not specifically estimate the cases of disease and deaths caused by SSB consumption (for example, of the 4,425 deaths mentioned, in the real world they are not only caused by SSB consumption but reducing SSB consumption may prevent them).

Thanks. We agree. We have now corrected it.

- Reformulating SSBs is not a good policy option to highlight in the abstract. I suggest discussing SSB taxation, marketing restrictions, or food and nutrition policies broadly.

Thanks. We agree. We have now corrected it.

Introduction

- The introduction is too long, and the structure makes it difficult to identify the issue being discussed. I suggest restructuring the introduction into three paragraphs:

1) Overweight, obesity and NCDs are a serious health concern globally.

2) An unhealthy diet, and consumption of SSBs is a key driver of overweight, obesity and NCDs. These are increasing worldwide.

3) In Argentina, consumption of SSBs is both high and increasing, and this is leading to an increased burden of disease.

4) Study rationale and aims

- Discussion of current policies to address unhealthy diets in Argentina should be left for the discussion.

Thanks. We agree. We have edited it following your suggestions.

Methods

- I feel that adding additional subheadings to the methods might clarify the approach for an audience with less economic modelling experience. For example, different sections for each of the three stages of estimating health and economic outcomes.

Thanks. We agree. We have edited it following your suggestions.

- Instead of a separate section on data sources, please describe each data source as its use is first mentioned in the text.

Thanks. We agree. We have edited it following your suggestions.

- Please define DALYs when first used

Thanks. We have now corrected it.

Results

- The main epidemiological and economic parameters included in the model should be presented in the methods, not the results (assuming that these are derived from pre-existing data sets).

Thanks. We have now corrected it following your suggestion.

- Please edit the results to use people-first language. For example, instead of “Overweight and obesity burden children and adolescents the most”, use “Children and adolescents face the greatest burden of overweight and obesity”.

Thanks. We have now edited it.

- Similarly, cases of obesity are not attributed to children/adolescents – children/adolescents are living with overweight and obesity.

Thanks. We have now edited it.

- In table 3, please present ‘events’ before ‘deaths’

Thanks. We have now edited it.

- It is difficult to say that SSBs cause 23% of all cases on diabetes, as so many factors contribute to disease. Instead, what the model estimates is that, if no SSBs were consumed, 23% of diabetes cases would be prevented. Please edit the results throughout to reflect this.

Thank you for your comment. We have now corrected it following your suggestion.

- I suggest not comparing the burden of disease across different NCDs as this implies that some are more important than others. Simply say how many cases of each would be avoided if SSB consumption was reduced to 0.

Thanks. We have now edited it following your suggestion.

- Please reference the results of the included sensitivity analyses

Thanks. We are now referring to sensitivity analysis findings.

Discussion

- As with the results, please be careful about directly attributing costs/deaths to SSB consumption – the model simply estimates that if SSB consumption were reduced, a certain number of costs and deaths would be prevented.

Thanks. We agree, and we have corrected it in this and other sections.

- Please mode the strengths and limitations to later in the discussion (just before the conclusion).

Thanks. We have now edited it following your suggestion.

- The authors state (in the limitations section) that their estimates are likely an underestimation, but it is not clear why this is the case – is this because oral health was not included? Please explain further.

Thanks. We have now edited it following your suggestion.

- Overall, the discussion would be improved by additional sections on drivers of SSB consumption and further consideration of policies. The authors discuss many policy options for addressing SSB consumption in Argentina, they may want to condense this a little and instead include more discussion on a) why SSB consumption is high (there is some discussion of advertising and prices, but there should be more on this) and b) how to implement best-practice policies (notably how to limit SSB industry influence).

Thanks. We have now added information concerning those points.

---

## [Decision Letter · Decision Letter 1]

10 Aug 2022

PONE-D-22-04375R1The Burden of Disease and Economic Impact of Sugar-Sweetened Beverages’ Consumption in Argentina: a Modeling StudyPLOS ONE

Dear Dr. Bardach,

Thank you for submitting your manuscript to PLOS ONE. After careful consideration, we feel that it has merit but does not fully meet PLOS ONE’s publication criteria as it currently stands. Therefore, we invite you to submit a revised version of the manuscript that addresses the points raised during the review process.

We look forward to receiving your revised manuscript.

Kind regards,

Vijay S. Gc, PhD

Academic Editor

PLOS ONE

Reviewers' comments:

Reviewer's Responses to Questions

**Comments to the Author**

1. If the authors have adequately addressed your comments raised in a previous round of review and you feel that this manuscript is now acceptable for publication, you may indicate that here to bypass the “Comments to the Author” section, enter your conflict of interest statement in the “Confidential to Editor” section, and submit your "Accept" recommendation.

Reviewer #1: (No Response)

Reviewer #3: All comments have been addressed

2. Is the manuscript technically sound, and do the data support the conclusions?

Reviewer #1: Yes

Reviewer #3: Partly

3. Has the statistical analysis been performed appropriately and rigorously? 

Reviewer #1: Yes

Reviewer #3: Yes

4. Have the authors made all data underlying the findings in their manuscript fully available?

Reviewer #1: No

Reviewer #3: Yes

5. Is the manuscript presented in an intelligible fashion and written in standard English?

Reviewer #1: No

Reviewer #3: Yes

6. Review Comments to the Author

Reviewer #1: The authors have done an excellent job of revising the manuscript in light of the suggestions from both reviewers. It flows well, it is clear and great to see the amended version. Congratulations.

A few further minor corrections are required for accuracy, consistency and transparency:

• It looks like there’s a few instances where writing in the present tense have slipped through and it should be past tense. Suggest doing a final check.

• I question your definition of a PIF. It looks like the definition of a PAF.

See:

Barendregt & Veerman (2009) Categorical versus continuous risk factors and the calculation of potential impact fractions. J Epidemiol Community Health. 64:209-212. doi:10.1136/jech.2009.090274

• Your sources of epidemiological parameters require references. What might be useful is including a technical appendix where all the data sources are provided in a table.

• Where did you get the definition of SSBs from? It needs a reference.

• You need to reference the exchange rates published by the Central Bank of the country. You have included it at the bottom of Table 2, it will be consistent to include it also in your text.

• In your text regarding economic values, clarify the values are US dollars (e.g., US$ 40 million, etc.) Otherwise, it is open to interpretation, even though you indicate in the tables it is USD. You could also just highlight at the beginning of your results all monetary values are in USD. Just make it clear.

• Something has gone wrong with ‘Deaths’ in table 3. Looks like an overlap or duplication.

• Discussion. Last sentence, third paragraph

What is the age group in numbers of the ‘youngest group’? The current definition tells the reader nothing. The reader is looking for detail (e.g.,” …..’incidence occurred in the youngest group (12-18 years), particularly among men……”

All the best for the final hurdle. It will be a great paper and will attract much interest from public health researchers and policy makers alike.

Reviewer #3: The consumption of sugar-sweetened beverages is a growing public health problem. In this study, the authors analyze SSB-associated burden in Argentina, one of the countries with the highest consumption of SSB in the world.

The authors have tried to carefully model the health burden and costs of SSB use. However, given the complexity of modeling itself, the article contains little information on the process of obtaining the results. I believe that the article needs a major revision of the methodological aspects before considering its publication in Plos One.

Here is a detail of suggested changes:

Major aspects

- reviewer #1 already mentioned the importance of providing accurate and explicit information about how the modeling process was conducted, and the lack of information in this regard. I do not believe you have addressed this concern. Although I understand that space constraints precludes authors from providing extensive and detailed data, these data are still needed, and you should consider adding an appendix that contains details about your assumptions.l. This appendix should at a minimum address/include:

- the list of all outcomes evaluated and their attached RRs (used in the PIF equation)

- a clarification of how the modeling pathway worked. For example:

- How exactly does BMI affect diabetes and CVD outcomes? (the link between direct and indirect pathways). Is not clear to me looking at Figure 1.

- How did you avoid duplicity of events? For example:

- you applied a risk of mortality attached to diabetes, but then another risk for CVD outcomes. Did you exclude the population of diabetics for the latest? If not, wouldn´t there be a duplicity of CVD events among diabetics?

- In figure 1, it seems you applied an effect of BMI on CVD (then reflects events through the direct pathway), but also a BMI effect directly on outcomes, which also includes CVD events. Did you differentiate between incident and established CVD?

- Are age and gender differences in results based solely on differences on SSB consumption? How do you account for the inherent differential risk of events among different age and gender groups?

- Cost analysis: although is not explicitly mentioned, it seems the analysis only covers one year of avoided events and related costs. Nevertheless, in table 2, you include a column describing costs since 2nd year/follow up. How were they modeled? Was there any assumption regarding economic burden in previous years?

- population size assumptions, by age and gender. Source of information for population inputs (like census)

- The paper models a scenario of not only a zero SSB consumption (as if they were withdrawn from market) but also no caloric compensation at all (an aspect that should be discussed at some point in the article, and it's currently not). Numbers are therefore strikingly high regarding SSB burden. Nevertheless, the discussion barely highlights the utopic nature of the modeled scenario, and I think it should be talked through in the discussion and highlighted in limitations.

Minor changes:

Introduction:

- 3rd paragraph, first sentence: it needs a reference.

- 3rd paragraph. The sentence “People who are overweight or obese are at greater risk of developing severe health problems, such as type 2 diabetes, high blood pressure, asthma, sleep disorders, liver disease, low self-esteem, and social and emotional issues” needs a reference

- Last paragraph reads: “This manuscript presents the results for Argentina obtained through a collaborative, more comprehensive project….”. More than what?

Methods:

- For what period of time is the intervention being conducted? It seems to be 2020 (for what it says in Results) but this should be clearly stated in Methods (and in the abstract)

- You used Monte Carlo simulations to get uncertainty intervals. I wouldn't call this a sensitivity analysis since you don't compare your results under a different assumption but rather estimate the uncertainty around the result you already got. In the same way, I wouldn´t call having reported UI a strength (as you do in discussion), since uncertainty around the main results is expected to be reported in almost any study.

Conclusion

- The last sentence reads: “This study is the first to assess the health and financial burden of SSB consumption in the country, including children and a wide range of attributable conditions.”. Nevertheless, and as you mentioned previously, the BASTA study already evaluated the health burden of SSB consumption among adults in Argentina. I believe this sentence is misleading and should be rephrased.

7. PLOS authors have the option to publish the peer review history of their article (what does this mean?). If published, this will include your full peer review and any attached files.

Reviewer #1: No

Reviewer #3: No

---

## [Author Response · Author response to Decision Letter 1]

15 Sep 2022

Reviewer #1: The authors have done an excellent job of revising the manuscript in light of the suggestions from both reviewers. It flows well, it is clear and great to see the amended version. Congratulations.

A few further minor corrections are required for accuracy, consistency and transparency:

• It looks like there’s a few instances where writing in the present tense have slipped through and it should be past tense. Suggest doing a final check.

Thanks. We have corrected these typos. 

• I question your definition of a PIF. It looks like the definition of a PAF.

See:

Barendregt & Veerman (2009) Categorical versus continuous risk factors and the calculation of potential impact fractions. J Epidemiol Community Health. 64:209-212. doi:10.1136/jech.2009.090274

Thanks for your comment. We analyzed both terms,1,2 and we agree that the best definition of our analysis is the PAF, so we modified it accordingly.

• Your sources of epidemiological parameters require references. What might be useful is including a technical appendix where all the data sources are provided in a table.

Thanks. We have already included all the epidemiological references in the manuscript. 

• Where did you get the definition of SSBs from? It needs a reference.

Thanks. We defined SSBs as the following ones: sports and energy drinks, sugar-sweetened sodas, sugar-sweetened fruit juices, and sugar-sweetened and flavored waters. This decision was based on a previous paper that assessed the disease burden attributed to SSBs (Singh, G. M., Micha, R., Khatibzadeh, S., Shi, P., Lim, S., Andrews, K. G., ... & Global Burden of Diseases Nutrition and Chronic Diseases Expert Group (NutriCoDE). (2015). Global, regional, and national consumption of sugar-sweetened beverages, fruit juices, and milk: a systematic assessment of beverage intake in 187 countries. PloS one, 10(8), e0124845.)

• You need to reference the exchange rates published by the Central Bank of the country. You have included it at the bottom of Table 2, it will be consistent to include it also in your text.

Thanks. We used the exchange rate published by the Central Argentinian Bank. We have corrected it within the manuscript. Also, now we are referencing it correctly in the manuscript.

• In your text regarding economic values, clarify the values are US dollars (e.g., US$ 40 million, etc.) Otherwise, it is open to interpretation, even though you indicate in the tables it is USD. You could also just highlight at the beginning of your results all monetary values are in USD. Just make it clear.

Thanks. Now we are just using one symbol, the $ symbol referring to USD. 

• Something has gone wrong with ‘Deaths’ in table 3. Looks like an overlap or duplication.

Thanks. We do not see that duplication in our document. Maybe it occurs when converting from word to pdf.

• Discussion. Last sentence, third paragraph

What is the age group in numbers of the ‘youngest group’? The current definition tells the reader nothing. The reader is looking for detail (e.g.,” …..’incidence occurred in the youngest group (12-18 years), particularly among men……”

Thanks. We have edited it according to your suggestions.

All the best for the final hurdle. It will be a great paper and will attract much interest from public health researchers and policy makers alike.

Many thanks. We really appreciate your suggestions.

Reviewer #3: The consumption of sugar-sweetened beverages is a growing public health problem. In this study, the authors analyze SSB-associated burden in Argentina, one of the countries with the highest consumption of SSB in the world.

The authors have tried to carefully model the health burden and costs of SSB use. However, given the complexity of modeling itself, the article contains little information on the process of obtaining the results. I believe that the article needs a major revision of the methodological aspects before considering its publication in Plos One.

Here is a detail of suggested changes:

Major aspects

- reviewer #1 already mentioned the importance of providing accurate and explicit information about how the modeling process was conducted, and the lack of information in this regard. I do not believe you have addressed this concern. Although I understand that space constraints preclude authors from providing extensive and detailed data, these data are still needed, and you should consider adding an appendix that contains details about your assumptions.l. This appendix should at a minimum address/include:

- the list of all outcomes evaluated and their attached RRs (used in the PIF equation)

Thanks. We are now including a statement and respective reference in the supplementary material with links to the specific GBD RRs used. 

- a clarification of how the modeling pathway worked. For example:

- How exactly does BMI affect diabetes and CVD outcomes? (the link between direct and indirect pathways). Is not clear to me looking at Figure 1.

Thanks for your comment. To avoid double counting for cardiovascular events, we just applied one RR value, which is the RR of the association between SSBs consumption and cardiovascular events (the same for diabetes). Since the RR of the association between SSBs consumption and cardiovascular events is not adjusted for BMI, somehow, the indirect association (through BMI) between BMI and cardiovascular (the same for diabetes) is also being considered. For that reason, we show in figure 1 two different arrows regarding the direct and indirect (through BMI) association between SSBs and cardiovascular events (the same for diabetes). We are explaining it better in the manuscript, and we modified Figure 1 to avoid a misunderstanding.

- How did you avoid duplicity of events? For example:

- you applied a risk of mortality attached to diabetes, but then another risk for CVD outcomes. Did you exclude the population of diabetics for the latest? If not, wouldn´t there be a duplicity of CVD events among diabetics?

Thanks. Your comment is really interesting. No, we are not double counting cardiovascular events. On the one hand, we assumed a RR value of 1.37 for diabetes type 2 incidence (assuming that it consequently affects prevalent cases and also mortality in the same way) per SSB serving day. 

On the other hand, we assumed a RR of 1.08 (95% CI, 1.04 to 1.13) and 1.08 (95% CI, 1.02 to 1.14) per serving per day for incidence and mortality of cardiovascular diseases. This value came from a meta-analysis in which the primary studies made adjustments for main potential confounders: 

“All studies adjusted for age, smoking, and physical activity (8–11, 13, 14, 16, 21, 23, 25, 48, 49). Most studies controlled for other RR factors, including BMI (8, 9, 11, 13, 14, 21, 23, 48, 49) (N = 9), education (8, 13, 14, 16, 23, 49) (N = 7), alcohol consumption (8, 9, 11, 13, 14, 16, 21, 23, 25, 48) (N = 10), total energy intake (8–11, 13, 14, 21, 23, 48, 49) (N = 10), and dietary quality (8–11, 14, 16, 21, 25, 48, 49) (N = 10). SSBs and LCSBs were mutually adjusted in 4 studies.” (Yin et al: https://www.ncbi.nlm.nih.gov/pmc/articles/PMC7850046/pdf/nmaa084.pdf)

As far as we know, these studies adjusted for main lifestyle potential confounders (many of which are also associated with diabetes developing) but did not adjust for diabetes particularly. However, our model did not count cardiovascular events attributed to diabetes, so we are not double-counting cardiovascular events. 

- In figure 1, it seems you applied an effect of BMI on CVD (then reflects events through the direct pathway), but also a BMI effect directly on outcomes, which also includes CVD events. Did you differentiate between incidents and established CVD?

Thank you for your comment. Yes, the model estimates incidents of CVD events attributed to BMI and how many established CVD events are also attributable to BMI.

- Are age and gender differences in results based solely on differences on SSB consumption? How do you account for the inherent differential risk of events among different age and gender groups?

Thank you for your comment. Most of the model inputs are differentiated by age and gender: BMI, SSB consumption, population, epidemiological data (incidence, prevalence, mortality), and the RR of BMI on the events (included in the supplementary material). Regarding the parameters of RR of SSB on BMI, CVD, or Diabetes, no information disaggregated by age and gender was found. We are adding this statement in the methods section now.

- Cost analysis: although is not explicitly mentioned, it seems the analysis only covers one year of avoided events and related costs. Nevertheless, in table 2, you include a column describing costs since 2nd year/follow up. How were they modeled? Was there any assumption regarding economic burden in previous years?

Thank you for your comment. As you have correctly mentioned, the model presents the diagnostic analysis of one year. The cost values that refer to consecutive years in Table 2 have been added to differentiate the costs in the incident and prevalent cases in those health conditions known to have essential differential costs between the first and consecutive years. So, we applied the healthcare costs only to incident cases and the consecutive year healthcare costs to the prevalent cases.

We added a explanation in the cost section:

“In turn, the costs were differentiated in the first year and subsequent years to distinguish and approximate the difference in costs of incident cases and prevalent cases.”

And we added a clarification in the model structure section:

“We used a comparative risk assessment framework to estimate the health and economic effects of sugar-sweetened beverage (SSB) consumption in one year.”

- population size assumptions, by age and gender. Source of information for population inputs (like census)

 Thanks. We have included this information in Table 1 (we use the last National Argentinian demographic source, as referenced)

- The paper models a scenario of not only a zero SSB consumption (as if they were withdrawn from market) but also no caloric compensation at all (an aspect that should be discussed at some point in the article, and it's currently not). Numbers are therefore strikingly high regarding SSB burden. Nevertheless, the discussion barely highlights the utopic nature of the modeled scenario, and I think it should be talked through in the discussion and highlighted in limitations.

Thanks. We totally agree with you. Now we are including a statement in the discussion section.

Minor changes:

Introduction:

- 3rd paragraph, first sentence: it needs a reference.

 Thanks. We have added a reference now.

- 3rd paragraph. The sentence “People who are overweight or obese are at greater risk of developing severe health problems, such as type 2 diabetes, high blood pressure, asthma, sleep disorders, liver disease, low self-esteem, and social and emotional issues” needs a reference

 Thanks. We have added a reference now.

- Last paragraph reads: “This manuscript presents the results for Argentina obtained through a collaborative, more comprehensive project….”. More than what?

Thanks. We have edited this sentence now.

Methods:

- For what period of time is the intervention being conducted? It seems to be 2020 (for what it says in Results) but this should be clearly stated in Methods (and in the abstract)

Thanks. We are mentioning this statement now. However, we would like to emphasize that there is no intervention here, since we are just estimating the disease and economic burden attributed to SSBs consumption. To do that, we are comparing the context in Argentina in 2020 with a hypothetical scenario of no SSB consumption.

- You used Monte Carlo simulations to get uncertainty intervals. I wouldn't call this a sensitivity analysis since you don't compare your results under a different assumption but rather estimate the uncertainty around the result you already got. In the same way, I wouldn´t call having reported UI a strength (as you do in discussion), since uncertainty around the main results is expected to be reported in almost any study.

Thanks. We have edited these sentences now.

Conclusion

- The last sentence reads: “This study is the first to assess the health and financial burden of SSB consumption in the country, including children and a wide range of attributable conditions.”. Nevertheless, and as you mentioned previously, the BASTA study already evaluated the health burden of SSB consumption among adults in Argentina. I believe this sentence is misleading and should be rephrased.

Thanks. We have edited this sentence now.

---

## [Decision Letter · Decision Letter 2]

18 Oct 2022

PONE-D-22-04375R2The Burden of Disease and Economic Impact of Sugar-Sweetened Beverages’ Consumption in Argentina: a Modeling StudyPLOS ONE

Dear Dr. Bardach,

Thank you for submitting your manuscript to PLOS ONE. After careful consideration, we feel that it has merit but does not fully meet PLOS ONE’s publication criteria as it currently stands. Therefore, we invite you to submit a revised version of the manuscript that addresses the points raised during the review process.

We look forward to receiving your revised manuscript.

Kind regards,

Vijay S. Gc, PhD

Academic Editor

PLOS ONE

Journal Requirements:

Reviewers' comments:

Reviewer's Responses to Questions

**Comments to the Author**

1. If the authors have adequately addressed your comments raised in a previous round of review and you feel that this manuscript is now acceptable for publication, you may indicate that here to bypass the “Comments to the Author” section, enter your conflict of interest statement in the “Confidential to Editor” section, and submit your "Accept" recommendation.

Reviewer #1: (No Response)

Reviewer #3: (No Response)

2. Is the manuscript technically sound, and do the data support the conclusions?

Reviewer #1: Yes

Reviewer #3: Yes

3. Has the statistical analysis been performed appropriately and rigorously? 

Reviewer #1: Yes

Reviewer #3: Yes

4. Have the authors made all data underlying the findings in their manuscript fully available?

Reviewer #1: Yes

Reviewer #3: Yes

5. Is the manuscript presented in an intelligible fashion and written in standard English?

Reviewer #1: Yes

Reviewer #3: Yes

6. Review Comments to the Author

Reviewer #1: Well done to the authors for improving this paper. There are only minor comments that require addressing:

Abstract

• If you have changed to describe a PAF, then you need to correct the wording in the abstract (it still says you’ve used the population impact factor).

Second page of introduction, fourth paragraph:

• The text indicates adults consume 85 litres of SSB per year. In the paragraph above that, it says children and adolescents consume 40% more SSBs that adults. Does that mean children and adolescents are consuming nearly 120litres per year?

Fourth page, first line:

• The authors need to correct the first line on this page to say…..’or mortality that would occur…….’

• Fourth line: the authors need to correctly say PAF, not population impact factor, and again, in the third paragraph on the same page.

Fifth page:

• Second paragraph: A reference needs to be included for the defining of SSBs

• Second paragraph from bottom: is it possible to superscript the BMI, like so: kg/m2?

Sixth page:

• Second paragraph, third line: it should read ….’per SSB serving per day

• Direct medical costs: the authors have not included in the methods section if the costs were calculated from a societal perspective or a health perspective. Is this relevant for this micro-costing approach (not knowing what a micro-costing approach is).

Table 3

• Although the authors have indicted there’s no problem with the ‘deaths’ row in Table 3, there still seems to be a doubling up of values over each other.

Sensitivity analysis

• In the second round of reviewing, reviewer 3 mentioned uncertainty analysis using bootstrapping is not sensitivity analysis. It appears this has not been corrected?

Figures – great graphs!

A general comment: The paragraphs spacing is inconsistent throughout. Please check.

Also, the authors use the word ‘information’ to describe data. Would it be more academic to describe it as data? Finally, it may be useful to get a final proofread - there are still typos.

Reviewer #3: Thank you for the opportunity to review this revised version of the manuscript.

We appreciate the changes the authors have made with regards to our previous comments, and appreciate the inclusion of the supplementary material.

Nevertheless, we've noticed that the list of all outcomes evaluated and their attached RRs (used in the PAF equation) is still incomplete (for example: there is still no information regarding how outcomes such as asthma, musculoskeletal conditions or other conditions were modeled. If you this this is to detailed, even for the appendix, please amend the answers “Yes - all data are fully available without restriction” and “All relevant data are within the manuscript”, and specify how interested people can contact you to request the missing information.

Regarding figure 1, if we are understanding it correctly, wouldn't something like this be more accurate, with CVD and other conditions as intermediate outcomes?

If we understood it incorrectly, please rearrange it so that it is clearer.

Finally, you say “Regarding the parameters of RR of SSB on BMI, CVD, or Diabetes, no

information disaggregated by age and gender was found”. We understand that RR may not be disaggregated by age and gender. We may have not been clear in our previous comment, but what we are wondering is: How did you model the risk of CVD and other diseases inherently associated with both age and gender, independent of the added risk of diabetes, BMI, or other variables under study? (For example, CVD risk is different for a 70-year-old man in comparison with a 35-year-old woman, even if both have no diabetes and a BMI below 25). Please clarify that in the methods.

On a minor note, you are still referring to Monte Carlo simulations as “Sensitivity analysis” (results section).

I think the manuscript has improved since the previous version but still needs these relatively minor clarifications. We look forward to your answers to our comments.

7. PLOS authors have the option to publish the peer review history of their article (what does this mean?). If published, this will include your full peer review and any attached files.

Reviewer #1: No

Reviewer #3: No

---

## [Author Response · Author response to Decision Letter 2]

3 Nov 2022

Deadline: Dec 02 2022 11:59P

Reviewer #1: Well done to the authors for improving this paper. There are only minor comments that require addressing:

Abstract

• If you have changed to describe a PAF, then you need to correct the wording in the abstract (it still says you’ve used the population impact factor).

Response: Thanks, we have edited it now.

Second page of introduction, fourth paragraph:

• The text indicates adults consume 85 litres of SSB per year. In the paragraph above that, it says children and adolescents consume 40% more SSBs that adults. Does that mean children and adolescents are consuming nearly 120litres per year?

Response: Yes. It can also be seen in Table 1.

Fourth page, first line:

• The authors need to correct the first line on this page to say…..’or mortality that would occur…….’

Response: Thanks, we have edited it now.

• Fourth line: the authors need to correctly say PAF, not population impact factor, and again, in the third paragraph on the same page.

Response: Thanks, we have edited it now.

Fifth page:

• Second paragraph: A reference needs to be included for the defining of SSBs

Response: Thanks, we have added it now.

• Second paragraph from bottom: is it possible to superscript the BMI, like so: kg/m2?

Response: Thanks, we have edited it now.

Sixth page:

• Second paragraph, third line: it should read ….’per SSB serving per day

Response: Thanks, we have edited it now.

• Direct medical costs: the authors have not included in the methods section if the costs were calculated from a societal perspective or a health perspective. Is this relevant for this micro-costing approach (not knowing what a micro-costing approach is).

Response: Thanks for this comment. We have clarified it now.

Table 3

• Although the authors have indicated there’s no problem with the ‘deaths’ row in Table 3, there still seems to be a doubling up of values over each other.

Response: Thanks. However, we do not see any overlap or duplication in Table 3 of our document. Maybe the Journal staff could help us with it.

Sensitivity analysis

• In the second round of reviewing, reviewer 3 mentioned uncertainty analysis using bootstrapping is not sensitivity analysis. It appears this has not been corrected?

Response: Thanks, we have edited it now.

Figures – great graphs!

A general comment: The paragraph's spacing is inconsistent throughout. Please check.

Also, the authors use the word ‘information’ to describe data. Would it be more academic to describe it as data? Finally, it may be useful to get a final proofread - there are still typos.

Response: Thanks, we appreciate those suggestions. We have edited it accordingly.

Reviewer #3: Thank you for the opportunity to review this revised version of the manuscript.

We appreciate the changes the authors have made with regards to our previous comments, and appreciate the inclusion of the supplementary material.

Nevertheless, we've noticed that the list of all outcomes evaluated and their attached RRs (used in the PAF equation) is still incomplete (for example: there is still no information regarding how outcomes such as asthma, musculoskeletal conditions or other conditions were modeled. If this is too detailed, even for the appendix, please amend the answers “Yes - all data are fully available without restriction” and “All relevant data are within the manuscript”, and specify how interested people can contact you to request the missing information.

Response: Many thanks. It is in our best interest that the methodology used in this analysis is well understood. Therefore, consequently to your suggestion, we have added detailed information in the supplementary appendix. Now all the formulas used can be seen, and you can understand better how we have modeled the indirect way (through BMI). Likewise, although the last PAF formula that we shared is mathematically correct and coincides with our analysis, we considered that it might be clearer for the reader to see it expressed in another way. That's why you'll see that we've also made this edit to the manuscript.

Regarding figure 1, if we are understanding it correctly, wouldn't something like this be more accurate, with CVD and other conditions as intermediate outcomes?

If we understood it incorrectly, please rearrange it so that it is clearer.

Response: Thanks for this comment. We are wondering if you tried to share with us some graphics since you mentioned, "wouldn't something like this be more accurate".

We will explain this more clearly so that it can be understood more adequately. Figure 1 is a general conceptual scheme of the model. It tries to summarize and simplifies what, in complex terms, the model estimates. The RR values of the direct effect between SSB consumption and incidence, prevalence, and mortality of Diabetes/CVD were Not reported as adjusted for BMI. For this reason, and to avoid double counting, we did not add RR values between BMI and incidence, prevalence and mortality of Diabetes/CVD. However, as the RR of the direct effect between SSB consumption and these two diseases was Not adjusted for BMI, our model somehow considers the effect between BMI and Diabetes/CVD. That is why Figure 1 schematically shows these two effect arrows reaching both Diabetes and CVD. Since the last figure was not fully understood, we are sending a new figure that, in our consideration, better represents what has been explained.

Finally, you say “Regarding the parameters of RR of SSB on BMI, CVD, or Diabetes, no

information disaggregated by age and gender was found”. We understand that RR may not be disaggregated by age and gender. We may have not been clear in our previous comment, but what we are wondering is: How did you model the risk of CVD and other diseases inherently associated with both age and gender, independent of the added risk of diabetes, BMI, or other variables under study? (For example, CVD risk is different for a 70-year-old man in comparison with a 35-year-old woman, even if both have no diabetes and a BMI below 25). Please clarify that in the methods.

Response: Thanks for this comment. This is correct. But our model does not need CVD risk information for every age and gender. Our model uses data on mortality and events by disease cause for all ages for the year 2020 in Argentina. Then, as the PAF formula explains, the model estimates the proportion of these deaths and events attributable to SSBs. Although the RR values of the consumption of SSB and CDV were the same for all genders and ages, the number of events and deaths from CDV was definitely different for the different ages and gender in Argentina in 2020. Also, the average consumption of SSB differed between those groups.

On a minor note, you are still referring to Monte Carlo simulations as “Sensitivity analysis” (results section).

Response: Thanks, we have edited it now.

I think the manuscript has improved since the previous version but still needs these relatively minor clarifications. We look forward to your answers to our comments.

Response: Many thanks. We really appreciate your suggestions.

---

## [Editor Report · Decision Letter 3]

20 Dec 2022

The Burden of Disease and Economic Impact of Sugar-Sweetened Beverages’ Consumption in Argentina: a Modeling Study

PONE-D-22-04375R3

Dear Dr. Bardach,

We’re pleased to inform you that your manuscript has been judged scientifically suitable for publication and will be formally accepted for publication once it meets all outstanding technical requirements.

Kind regards,

Vijay S. Gc, PhD

Academic Editor

PLOS ONE

Additional Editor Comments (optional):

Please correct the first sentence under the Results section in the abstract. 110,000 healthy life-years (DALYs) to 110,000 healthy life years lost to premature death and disability.
---

## [Editor Report · Acceptance letter]

26 Dec 2022

PONE-D-22-04375R3 

The Burden of Disease and Economic Impact of Sugar-Sweetened Beverages' Consumption in Argentina: A Modeling Study. 

Dear Dr. Bardach:

I'm pleased to inform you that your manuscript has been deemed suitable for publication in PLOS ONE. Congratulations! Your manuscript is now with our production department. 

Kind regards, 

on behalf of

Dr. Vijay S. Gc 

Academic Editor

PLOS ONE